# GRADIENT-DIRECTION-AWARE DENSITY CONTROL FOR 3D GAUSSIAN SPLATTING

**Zheng Zhou[1], Yu-Jie Xiong[1,*], Jia-Chen Zhang[1], Chun-Ming Xia[1], Xihe Qiu[1], Hongjian Zhan[2]**
[1]Shanghai University of Engineering Science, Shanghai, China
[2]East China Normal University, Shanghai, China
`{m320123332, xiong, m325123603, cmxia}@sues.edu.cn`
`qiuxihe1993@gmail.com, hjzhan@cee.ecnu.edu.cn`

## ABSTRACT

The emergence of 3D Gaussian Splatting (3DGS) has significantly advanced Novel View Synthesis (NVS) through explicit scene representation, enabling real-time photorealistic rendering. However, existing approaches manifest two critical limitations in complex scenarios: (1) Over-reconstruction occurs when persistent large Gaussians cannot meet adaptive splitting thresholds during density control. This is exacerbated by conflicting gradient directions that prevent effective splitting of these Gaussians; (2) Over-densification of Gaussians occurs in regions with aligned gradient aggregation, leading to redundant component proliferation. This redundancy significantly increases memory overhead due to unnecessary data retention. We present Gradient-Direction-Aware Gaussian Splatting (GDAGS) to address these challenges. Our key innovations: the Gradient Coherence Ratio (GCR), computed through normalized gradient vector norms, which explicitly discriminates Gaussians with concordant versus conflicting gradient directions; and a nonlinear dynamic weighting mechanism leverages the GCR to enable gradient-direction-aware density control. Specifically, GDAGS prioritizes conflicting-gradient Gaussians during splitting operations to enhance geometric details while suppressing redundant concordant-direction Gaussians. Conversely, in cloning processes, GDAGS promotes concordant-direction Gaussian densification for structural completion while preventing conflicting-direction Gaussian overpopulation. Comprehensive evaluations across diverse real-world benchmarks demonstrate that GDAGS achieves superior rendering quality while effectively mitigating over-reconstruction, suppressing over-densification, and constructing compact scene representations. Our code is available at `https://github.com/zzcqz/GDAGS`.

## 1 INTRODUCTION

Novel View Synthesis (NVS) constitutes a foundational challenge in computer graphics and vision, aiming to reconstruct 3D scenes from monocular images to generate photorealistic novel perspectives. While neural implicit representations, such as Neural Radiance Fields (NeRF) (Mildenhall et al., 2021), have significantly advanced NVS by modeling radiation fields through multilayer perceptrons (MLPs) and volumetric rendering, these methods struggle to reconcile rendering quality (Verbin et al., 2022) with computational efficiency (Fridovich-Keil et al., 2022). Emerging as a novel paradigm, 3D Gaussian Splatting (3DGS) (Kerbl et al., 2023) explicitly parameterizes scenes using adaptive Gaussian primitives. By integrating splat-based rasterization with adaptive density control strategies, 3DGS achieves real-time rendering of unprecedented fidelity while maintaining compact spatial representation.

However, the adaptive density control mechanism in 3DGS exhibits inherent limitations (Ye et al., 2024; Huang et al., 2025; Zeng et al., 2025). Specifically, 3DGS's densification criterion solely relies on the norm of view-space positional gradients, thereby overlooking the directional influence of gradients on the norm computation. During view-space positional gradient calculation for each

---
*Corresponding author.

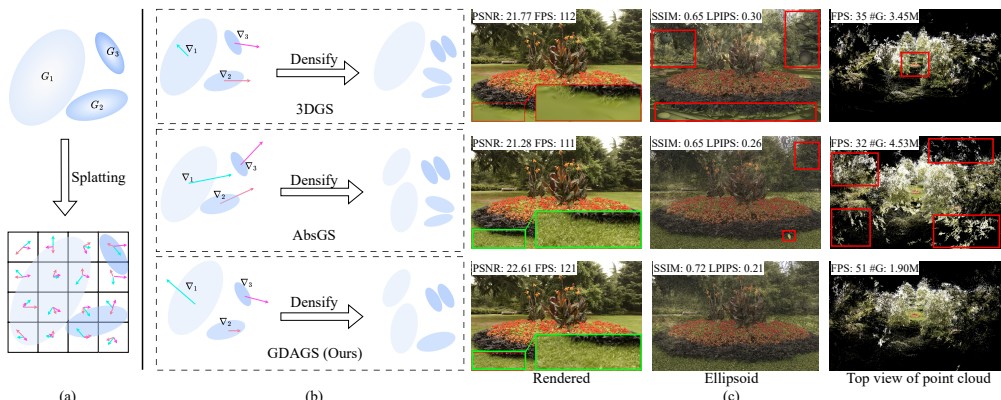

Figure 1: (a) illustrates the Gaussian ellipsoid splatting process, where arrows of different colors represent the gradient direction and magnitude of different Gaussians on the pixels. (b) shows the densification process of different methods. In 3DGS, a large Gaussian covering many pixels may fail to split because the combined gradient magnitude from different pixels falls below the threshold, leading to over-reconstruction as shown in the Rendered part of (c), which manifests as blurry areas. AbsGS forces all Gaussian gradients to be positive, causing the combined gradient magnitude from different pixels to increase significantly. This results in a substantial rise in the number of splitting Gaussians, leading to over-densification as shown in the Top view of the point cloud part of (c), which appears as a large number of Gaussian point clouds outside the scene. Our proposed method utilizes gradient direction information to effectively control the splitting of large Gaussians and the growth in the number of small and medium Gaussians. (c) also presents the performance (performance metrics are labeled in the top-left corner) and efficiency (FPS and total number of Gaussians in the scene, #G) of different methods.

Gaussian, the superposition of sub-gradient components with consistent directions amplifies the gradient norm. Conversely, counter-directional sub-gradient components mutually attenuate, resulting in reduced gradient norms. These directional conflicts prevent some Gaussians from effective densification, causing the over-reconstruction phenomenon and perceptible local blurring (Zhang et al., 2024b; Zhou et al., 2025). Meanwhile, Gaussians maintaining consistent gradient directions exhibit persistent densification tendencies, causing over-densification phenomena that generate redundant Gaussian primitives and significantly increase memory consumption. Recent work like AbsGS (Ye et al., 2024) has proposed using absolute subgradient values to enforce positive gradients for all Gaussians, thereby avoiding subgradient directional conflicts and partially mitigating the over-reconstruction issue, but this approach further exacerbates the over-densification phenomenon.

To address the aforementioned dual challenges, we propose a gradient-direction-aware adaptive density control framework (GDAGS). This approach combines directional gradient information and gradient magnitude data to establish a novel decision metric. The metric enables controlled densification of Gaussians with conflicting directional gradients while preventing redundant densification in directionally aligned cases. First, GDAGS calculates GCR to measure the directional consistency of subgradients for each Gaussian. The GCR approaching 1 indicates highly consistent subgradient directions, whereas a value approaching 0 signifies substantial directional conflict. Subsequently, GDAGS employs GCR to parameterize a nonlinear dynamic weight $w_i$, which independently weights the view-space positional gradients of each Gaussian to form a new decision metric controlling densification behavior. During splitting operations, Gaussians exhibiting directional conflicts receive higher weights and are prioritized for splitting, while directionally consistent Gaussians obtain lower weights and are suppressed. Finally, GDAGS introduces a hyperparameter $p$ to explicitly control the number of Gaussians participating in densification, thereby achieving an indirect balance between performance and efficiency. Comparative results and efficiency analyses are summarized in Figure 1. We summarize our contributions as follows:

- We propose GCR as a gradient direction consistency measure for adaptive density control to identify Gaussians with consistent and divergent directions.

- We propose a nonlinear dynamic weighting mechanism to independently weight the Gaussians in the scene, promoting correct densification and suppressing abnormal densification of the Gaussians.
- Evaluated on three real-world datasets spanning 13 scenes, our method demonstrates superior visual quality while having a compact spatial representation.

## 2 RELATED WORK

### 2.1 NOVEL VIEW SYNTHESIS

Novel View Synthesis has evolved through two interconnected trajectories: advancements in scene representation paradigms and optimizations of rendering efficiency. The field was revolutionized by NeRF (Mildenhall et al., 2021), which introduced MLP-based implicit neural representations to map spatial coordinates to radiance and density values, achieving photorealistic results through volumetric rendering. However, NeRF's computational demands spurred research bifurcating into quality refinement and efficiency optimization. Approaches like Mip-NeRF (Barron et al., 2021) enhanced geometric stability using conical ray encoding and distortion regularization, while DVGO (Sun et al., 2022) and Instant-NGP (Müller et al., 2022) leveraged explicit 3D grids and hash encoding to achieve near-real-time rendering. This progression culminates in 3D Gaussian Splatting (Kerbl et al., 2023), which redefined the paradigm by combining explicit Gaussian primitives with differentiable rasterization. By parameterizing scenes as adaptive 3D Gaussians, 3DGS bridged the gap between implicit continuity and explicit computational efficiency, achieving rendering speeds two orders of magnitude faster than NeRF while maintaining visual parity with state-of-the-art methods like Mip-NeRF360 (Barron et al., 2022).

### 2.2 3D GAUSSIAN SPLATTING

The rapid adoption of 3DGS has driven innovations addressing its inherent limitations (Cheng et al., 2024; Hamdi et al., 2024; Kheradmand et al., 2024; Lee et al., 2024; Li et al., 2025b). Early improvements focus on geometric fidelity: Mip-Splatting (Yu et al., 2024a) integrates multi-scale primitives and low-pass filtering to mitigate aliasing, while GOF (Yu et al., 2024b) introduces composite gradient metrics to optimize Gaussian distributions. Subsequent methods explore structural constraints, such as 2DGSs (Huang et al., 2024) planar surface projections and RadeGSs (Zhang et al., 2024a) normal map-guided refinement. Frequency-aware approaches like FreGS (Zhang et al., 2024b) employ spectral regularization to enhance high-frequency detail recovery, whereas HoGS (Liu et al., 2025) utilizes homogeneous coordinate reparameterization to improve distant object reconstruction. Despite these advances, critical challenges persist in adaptive density control. AbsGS (Ye et al., 2024) enforces gradient direction uniformity to resolve over-reconstruction but indiscriminately amplifies outliers, while PixelGS (Zhang et al., 2024c) trades memory efficiency for geometric precision by weighting gradients via pixel coverage—a strategy incurring prohibitive storage costs and frame rate penalties. ReAct-GS (Cheng et al., 2025) proposes a densification and reactivation mechanism based on importance perception to improve rendering quality, while PSRGS (Li et al., 2025a) solves the problem of high-frequency detail loss by identifying geometric and texture coupling regions. These methods inevitably increase model complexity by introducing additional modules or strategies. Our work addresses these unresolved trade-offs through directional gradient awareness, systematically balancing precision, efficiency, and memory constraints.

## 3 METHOD

Section 3.1 establishes the methodological motivation for GDAGS, and Section 3.2 outlines 3DGS fundamentals. Sections 3.3 detail GDAGS' core components, with full algorithmic implementation provided in Appendix A. The complete pipeline is shown in Figure 2.

### 3.1 PRELIMINARY

3DGS represents scenes using a set of learnable anisotropic 3D Gaussian primitives. Each Gaussian is parameterized by a position $\mu \in \mathbb{R}^3$, an opacity $\alpha \in [0, 1]$, a covariance matrix $\Sigma \in \mathbb{R}^{3 \times 3}$,

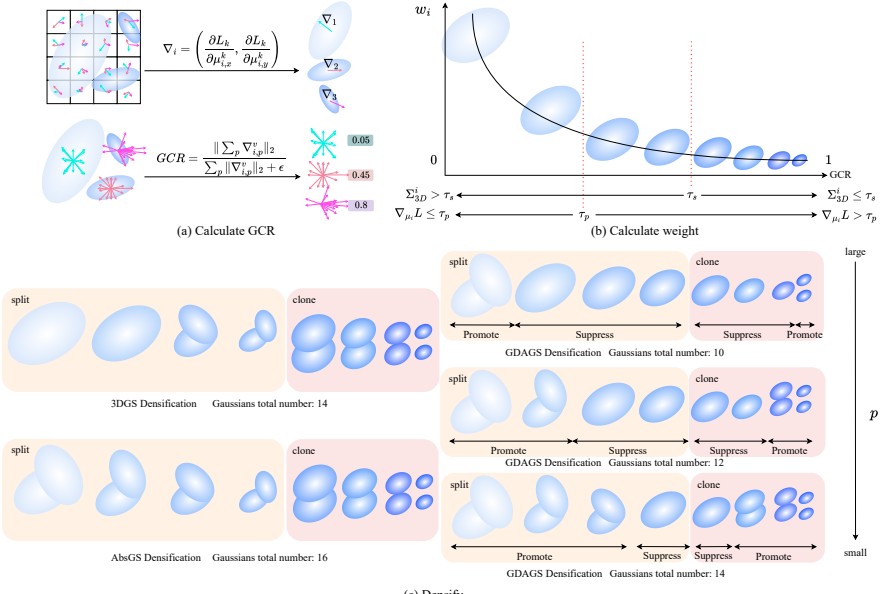

Figure 2: The pipeline of GDAGS. First, for each Gaussian, GDAGS computes the GCR to quantify the directional coherence of its subgradients. Subsequently, this GCR metric is mapped through a nonlinear dynamic weighting function to generate per-Gaussian gradient weights, which modulate the view-space positional gradient magnitudes and produce a refined decision metric. Finally, this decision metric is compared against a predefined threshold to dynamically regulate densification.

and Spherical Harmonics (SH) (Arfken et al., 2011) coefficients for view-dependent color. The covariance matrix $\Sigma$ is decomposed into a rotation matrix $R$ and a scaling matrix $S$ to ensure positive semi-definiteness:

$$\Sigma = RSS^\top R^\top, \tag{1}$$

where $S$ is a diagonal matrix parameterized by a vector $s \in (\mathbb{R}_+)^3$, and $R$ is a rotation matrix parameterized by a quaternion $q \in \mathbb{R}^4$. The Gaussian distribution is defined as:

$$G(\mathbf{x}) = \exp\left(-\frac{1}{2}(\mathbf{x} - \mu)^\top \Sigma^{-1}(\mathbf{x} - \mu)\right), \tag{2}$$

where $\mathbf{x} \in \mathbb{R}^3$ is an arbitrary position.

During rendering, 3D Gaussians are projected into 2D camera space as $G'$, with pixel color $C(\mathbf{x}')$ computed via alpha-blending:

$$C(\mathbf{x}') = \sum_{k=1}^{K} c_k \alpha_k G'k(\mathbf{x}')T_k, \quad T_k = \prod_{j=1}^{k-1}\left(1 - \alpha_j G'_j(\mathbf{x}')\right), \tag{3}$$

Here, $K$ denotes the count of Gaussians influencing the pixel, $c_k$ represents the view-dependent SH-based color, and $\alpha_k$ combines opacity with the Gaussian's spatial contribution at $\mathbf{x}'$.

To address sparse input from structure-from-motion (SfM) (Schonberger & Frahm, 2016), a densification strategy is employed. Under-reconstructed regions (missing geometric features) are addressed by cloning Gaussians, increasing their number and volume. Over-reconstructed regions (large Gaussians covering small areas) are split into smaller Gaussians, maintaining volume but increasing count. Densification decisions are based on view-space gradients. For a Gaussian $G_i$ with projection $\mu_i^k$ under viewpoint $k$ and loss $L_k$, the average gradient magnitude is:

$$\nabla_{\mu_i} L = \frac{1}{M}\sum_{k=1}^{M}\sqrt{\left(\frac{\partial L_k}{\partial \mu_{i,x}^k}\right)^2 + \left(\frac{\partial L_k}{\partial \mu_{i,y}^k}\right)^2}, \tag{4}$$

where $M$ is the number of viewpoints involved in training.

Split Gaussians into smaller components if $\nabla_{\mu_i} L > \tau_p$ and $\Sigma^i_{3D} > \tau_s$. Clone Gaussians to preserve local density if $\nabla_{\mu_i} L > \tau_p$ and $\Sigma^i_{3D} \leq \tau_s$.

## 3.2 MOTIVATION

The density control in 3D Gaussian Splatting primarily depends on comparing the norms of view-space positional gradients against predefined thresholds. However, the directional properties of these gradients implicitly influence their magnitudes—either attenuating or amplifying them—leading to undesirable effects such as local blurring and excessive Gaussian proliferation, which ultimately impair rendering quality. To overcome these issues, we introduce a novel decision metric that explicitly incorporates subgradient direction through a nonlinear weighting function, effectively addressing both over-reconstruction and over-densification. Specifically, during splitting operations—which target larger Gaussians—we use the GCR to guide the weighting strategy. Gaussians with smaller GCR values, indicating directional conflicts, receive an amplified decision metric via nonlinear weighting, thus prioritizing their densification. In contrast, those with larger GCR values, reflecting directional consistency, are assigned a diminished metric and suppressed. During cloning operations, which affect smaller Gaussians, the logic is reversed: low-GCR Gaussians are suppressed, while high-GCR Gaussians are promoted. This is because high-GCR clones can smoothly propagate along the gradient direction, whereas low-GCR clones tend to accumulate locally without meaningful displacement. By systematically integrating both gradient direction and magnitude into the densification criterion, GDAGS optimizes the density control mechanism. This approach not only mitigates key limitations of standard 3DGS but also significantly reduces the number of Gaussians required and corresponding memory usage.

## 3.3 GDAGS

### 3.3.1 DIRECTIONAL CONSISTENCY MEASUREMENT

For each Gaussian observed across $V$ views, we compute directional consistency metric GCR as $\mathcal{C}_i$:

$$\mathcal{C}_i = \frac{\| \sum_{pixel} \nabla^v_{i,pixel} \|_2}{\sum_{pixel} \| \nabla^v_{i,pixel} \|_2 + \epsilon}, \tag{5}$$

where $pixel$ denotes a pixel in the 2D space, and $\nabla^v_{i,pixel} \in \mathbb{R}^2$ represents the subgradient component projected onto each pixel in view $v$. According to the Cauchy-Schwarz Inequality $0 \leq \| \sum_{pixel} \nabla^v_{i,pixel} \|_2 \leq \sum_{pixel} \| \nabla^v_{i,pixel} \|_2$, it can be seen that the value of $\mathcal{C}_i$ is strictly controlled between [0,1], and GCR isolates the influence of gradient amplitude and only represents the consistency of gradient direction. This formulation captures:

- **High Consistency** ($\mathcal{C}_i \to 1$): Gradient directions are consistent, their magnitudes effectively accumulate.
- **Low Consistency** ($\mathcal{C}_i \to 0$): Conflicting gradient directions result in mutual cancellation of magnitudes.

The denominator $\sum_{pixel} \| \nabla^v_{i,pixel} \|_2$ captures the total gradient activity agnostic to directional variations, while the numerator $\| \sum_{pixel} \nabla^v_{i,pixel} \|_2$ quantifies the net directional alignment. Due to the fact that GCR calculates subgradients for each pixel in multiple views, GDAGS works under a clear assumption that most Gaussian distributions in the scene can receive sufficient gradient information.

### 3.3.2 NONLINEAR DYNAMIC WEIGHT DESIGN

The core of our adaptive density control lies in transforming the GCR into a dynamic weight that modulates the gradient magnitude used in densification decisions. A naive linear weighting scheme proves insufficient, as it lacks the expressive power to aggressively suppress high-consistency Gaussians while sensitively amplifying those with critical directional conflicts. To address this, we

introduce a nonlinear weighting function designed with two key objectives: strongly attenuating the contribution of Gaussians with high GCR values to prevent redundant densification in well-defined regions. And strategically amplify the influence of Gaussians with intermediate to low GCR values, prioritizing them for densification to resolve directional conflicts and enhance geometric detail. We design a nonlinear weighting function defined as:

$$w_i = \alpha + \beta \cdot (1 - \mathcal{C}_i)^p, \tag{6}$$

where $\mathcal{C}_i$ is the gradient coherence ratio for Gaussian $i$ computed via Equation 5, $\alpha$ is the fundamental inhibitory factor that suppresses Gaussian densification with consistent gradient directions during the splitting process, and suppresses Gaussian densification with inconsistent gradient directions during the cloning process. $\beta$ acts as the amplification factor, scaling the weight for Gaussians with strong directional conflicts, and $p$ governs the steepness of the exponential suppression, rapidly decreasing the weight for Gaussians with high directional consistency.

The exponential form is motivated by the need for a strong, monotonic decay that effectively neutralizes the influence of Gaussians whose gradients are already aligned. This prevents the unnecessary splitting or cloning that leads to over-densification. The power-law form in the amplification region, $(1 - \mathcal{C}_i)^p$, is chosen because it is more sensitive to changes in $\mathcal{C}_i$ when consistency is low compared to a linear function. This sensitivity is crucial for identifying and prioritizing Gaussians that are large and cover areas with complex geometry (and hence conflicting gradients), which are most in need of splitting.

The modulated gradient norm for Gaussian $i$ is then computed as:

$$\tilde{\nabla}_{\mu_i} L = w_i \cdot \nabla_{\mu_i} L. \tag{7}$$

This refined metric replaces the original gradient norm in all densification decisions, enabling consistent gradient-direction-aware control during optimization. The weighting function is applied strategically according to the densification operation: during splitting, the weight $w_i$ is applied directly to prioritize the fragmentation of large Gaussians exhibiting directional conflicts. In cloning operations, the inverse policy $(1/w_i)$ is adopted to encourage the propagation of small Gaussians with high GCR along surface directions, while suppressing those with low GCR to prevent chaotic local accumulation. Comprehensive ablation studies in Section 4.3.1 confirm the superiority of this nonlinear weighting strategy over linear alternatives, and a sensitivity analysis of parameters $p$ and $\beta$ is provided in Section 4.3.2.

## 4 EXPERIMENTS

### 4.1 SETUP

**Datasets.** Similar to the 3DGS dataset, we select all 9 scenes introduced in MIP-NERF360 (Barron et al., 2022), including indoor and outdoor scenes, Train and Truck scenes in Tanks&Temples (Hedman et al., 2018), and drJohnson and playroom scenes in Deep Blending (Knapitsch et al., 2017).

**Baselines.** The proposed method is benchmarked against three categories of approaches: fast NeRF frameworks (Plenoxels (Fridovich-Keil et al., 2022), Instant-NGP (Müller et al., 2022)), state-of-the-art NeRF-based models (Mip-NeRF360 (Barron et al., 2022)), and 3DGS variants including the widely adopted 3DGS (Kerbl et al., 2023), Taming 3DGS (Mallick et al., 2024) and mini-splatting (Fang & Wang, 2024), over-reconstruction-focused AbsGS (Ye et al., 2024), and Pixel-GS (Zhang et al., 2024c), ensuring comprehensive evaluation across all datasets.

**Implementation Details.** Our experimental setup follows the original paper parameter settings of 3DGS, with densification performed every 100 iterations, stopping densification after 15k iterations and completing training at 30k iterations. The suppression factor $\alpha$ in the weight function is set to 0.8, the amplification factor $\beta$ is set to the size threshold of 25, and the hyperparameter $p$ is set to 15. All experiments are conducted on a single NVIDIA 4090 GPU with 24GB of memory. We report the view metrics PSNR, SSIM, and LPIPS (Zhang et al., 2018), and the memory used to store Gaussian parameters to demonstrate the trade-off between performance and efficiency.

Table 1: Quantitative comparison on three datasets. SSIM↑ and PSNR↑ are higher-the-better; LPIPS↓ is lower-the-better. For fair comparison and to balance the trade-off between overall quality and memory consumption, we train these datasets with the same settings as 3DGS. All methods use the same training data for training. The best score , second best score are red and orange, respectively.

| Datasets | Mip-NeRF360 | | | | Tanks&Temples | | | | Deep Blending | | | |
|---|---|---|---|---|---|---|---|---|---|---|---|---|
| Methods | SSIM↑ | PSNR↑ | LPIPS↓ | Mem↓ | SSIM↑ | PSNR↑ | LPIPS↓ | Mem↓ | SSIM↑ | PSNR↑ | LPIPS↓ | Mem↓ |
| Plenoxels | 0.626 | 23.08 | 0.463 | 2.1GB | 0.719 | 21.08 | 0.379 | 2.3GB | 0.795 | 23.06 | 0.510 | 2.7GB |
| INGP | 0.671 | 25.30 | 0.371 | 13MB | 0.723 | 21.72 | 0.330 | 13MB | 0.797 | 23.62 | 0.423 | 13MB |
| Mip-NeRF360 | 0.792 | 27.69 | 0.237 | 8.6MB | 0.759 | 22.22 | 0.257 | 8.6MB | 0.901 | 29.40 | 0.245 | 8.6MB |
| 3DGS | 0.815 | 27.21 | 0.214 | 734MB | 0.841 | 23.14 | 0.183 | 411MB | 0.903 | 29.41 | 0.243 | 676MB |
| Pixel-GS | 0.832 | 27.72 | 0.178 | 1.2GB | 0.853 | 23.74 | 0.150 | 1.05GB | 0.896 | 28.91 | 0.248 | 1.1GB |
| AbsGS | 0.820 | 27.49 | 0.191 | 728MB | 0.853 | 23.73 | 0.162 | 304MB | 0.902 | 29.67 | 0.236 | 444MB |
| Taming 3DGS | 0.822 | 27.79 | 0.205 | 735MB | 0.851 | 24.04 | 0.170 | 411MB | 0.907 | 30.14 | 0.235 | 676MB |
| mini-splatting | 0.822 | 27.34 | 0.217 | 225MB | 0.835 | 23.18 | 0.202 | 130MB | 0.908 | 29.98 | 0.253 | 70MB |
| **GDAGS (Ours)** | 0.839 | 28.02 | 0.145 | 515MB | 0.854 | 23.79 | 0.165 | 226MB | 0.905 | 29.70 | 0.235 | 388MB |

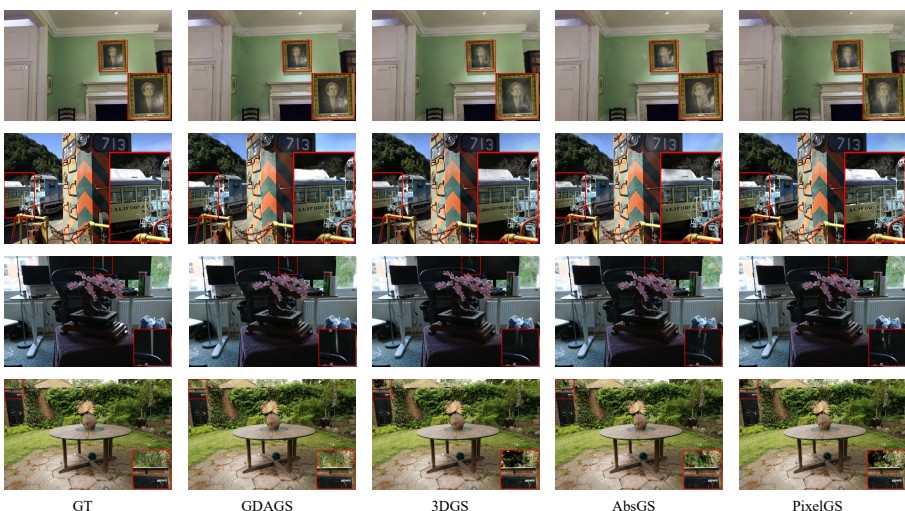

GT   GDAGS   3DGS   AbsGS   PixelGS

Figure 3: Qualitative comparisons of different methods on scenes from Mip-NeRF360, Tanks&Temples and Deep Blending datasets. Enlarged images are displayed in the bottom right corner.

## 4.2 COMPARISON

**Quantitative Results.** We report the quantitative results of the experiment in the Table 1, which indicate that our method produces comparable results in most cases. It is worth noting that our method has less memory at performance close to or better than SOTA. Compared to Pixel-GS, GDAGS achieves comparable or superior rendering performance with only 20%-50% of the memory consumption. This, combined with its comparison against 3DGS, demonstrates that GDAGS effectively addresses over-densification through nonlinear dynamic weights that suppress redundant Gaussians during densification. By adaptively controlling density, GDAGS enables scene Gaussians to converge toward compact spatial representations.

Compared to AbsGS, which enforces non-negative gradient components to resolve directional conflicts, this approach introduces a new issue: it amplifies the influence of outlier gradients arising from noisy or suboptimal regions. As a result, it triggers excessive and often unnecessary splitting, leading to over-densification, higher memory consumption, and noisier renderings—without a proportional improvement in visual quality. In contrast, Pixel-GS employs a pixel-coverage weighting mechanism that enhances spatial adaptation. However, it fails to account for directional information in gradients. This omission means it still encourages densification in areas with large yet directionally consistent gradients, resulting in redundant Gaussians and elevated memory costs. Our method addresses these limitations by leveraging directional coherence to actively suppress such redundant densification.

**Qualitative Results.** We present NVS results of GDAGS and baseline methods in Figure 3, demonstrating that our approach achieves high-quality visual fidelity across all scenes while effectively mitigating localized blur and detail loss. Comprehensive quantitative and qualitative results are presented in the Appendix D.

## 4.3 ANALYSIS

Table 2: Ablation experiment on three datasets. SSIM↑ and PSNR↑ are higher-the-better; LPIPS↓ is lower-the-better. The  best score , and  second best score  are red, and orange, respectively.

| Datasets | Mip-NeRF360 | | | | Tanks&Temples | | | | Deep Blending | | | |
|---|---|---|---|---|---|---|---|---|---|---|---|---|
| Methods | SSIM↑ | PSNR↑ | LPIPS↓ | Mem↓ | SSIM↑ | PSNR↑ | LPIPS↓ | Mem↓ | SSIM↑ | PSNR↑ | LPIPS↓ | Mem↓ |
| 3DGS | 0.815 | 27.21 | 0.214 | 734MB | 0.841 | 23.14 | 0.183 | 411MB | 0.903 | 29.41 | 0.243 | 676MB |
| GDAGS-L | 0.814 | 27.55 | 0.248 | 713MB | 0.849 | 23.74 | 0.179 | 321MB | 0.893 | 29.62 | 0.241 | 397MB |
| GDAGS-S | 0.819 | 27.52 | 0.240 | 441MB | 0.846 | 23.54 | 0.178 | 195MB | 0.905 | 29.67 | 0.238 | 328MB |
| GDAGS-C | 0.812 | 27.46 | 0.217 | 615MB | 0.847 | 23.71 | 0.180 | 305MB | 0.904 | 29.70 | 0.240 | 452MB |
| **GDAGS (Ours)** | 0.839 | 28.02 | 0.145 | 515MB | 0.852 | 23.79 | 0.165 | 226MB | 0.905 | 29.70 | 0.235 | 388MB |

### 4.3.1 ABLATION STUDY

We perform ablation studies on the proposed weighting strategy by training three model variants: GDAGS-S (applying weighting only during splitting), GDAGS-C (only during cloning), and GDAGS-L (using a linear weighting function $w_i = 2 - C_i$). These are compared against the original 3DGS and the full GDAGS model. Results are summarized in Table 2. The study reveals that split-oriented weighting (GDAGS-S) effectively improves SSIM and LPIPS while reducing memory usage. Clone-oriented weighting (GDAGS-C) enhances reconstruction fidelity (PSNR) but at the cost of increased memory consumption. Integrating both strategies in the full GDAGS model achieves an optimal balance between performance and efficiency. Notably, the nonlinear weighting function in GDAGS significantly outperforms its linear counterpart (GDAGS-L), as it more effectively translates gradient direction coherence into adaptive control signals. This nonlinear mapping provides superior sensitivity to directional conflicts and consistency, leading to more stable optimization and higher-quality results across all metrics compared to the linear approximation.

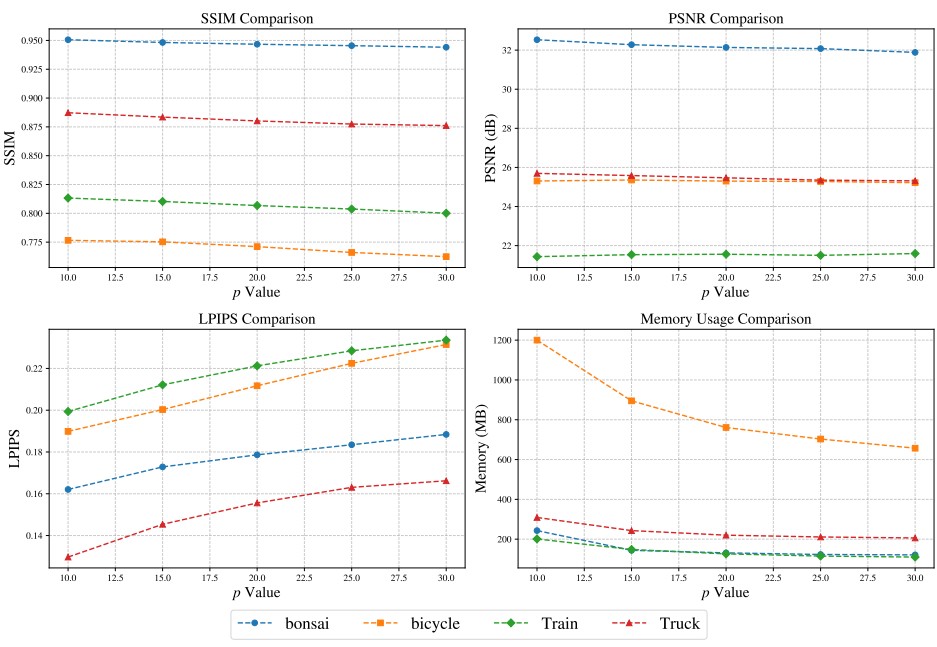

Figure 4: Performance of different hyperparameters $p$ in multiple datasets.

### 4.3.2 PARAMETER SENSITIVITY ANALYSIS

A sensitivity analysis is performed on hyperparameters $p$ and $\beta$, with detailed results for $\beta$ included in the Appendix C. The study shows that $p$ exerts a stronger influence on both rendering performance and computational efficiency. As demonstrated in Figure 4, we test $p$ across a range of values: 10, 15, 20, 25, and 30. As previously explained, $p$ determines the proportion of Gaussians selected for densification within the adaptive control mechanism. Increasing $p$ restricts densification to fewer Gaussians, which reduces memory usage but also compromises rendering quality. In contrast, a smaller $p$ expands the candidate set for densification, enhancing visual performance at the expense of greater memory overhead. These findings underscore the critical role of $p$ in balancing the trade-off between efficiency and reconstruction quality.

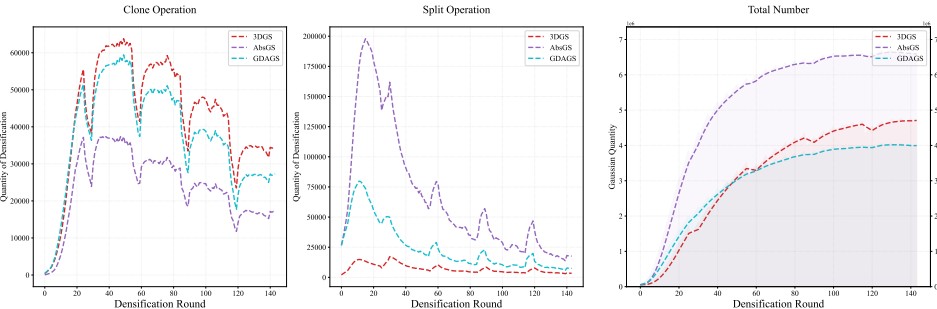

Figure 5: Visualization of different densification methods during the training process in bicycle sense.

### 4.3.3 DENSIFICATION PROCESS ANALYSIS

We quantified the Gaussian split/clone counts per densification round and the total Gaussian population dynamics during training in Figure 5. The experimental results demonstrate distinct behavioral patterns: In cloning operations, all three methods exhibit consistent trends, with 3DGS achieving the highest clone count due to its standard threshold. AbsGS, employing twice the cloning threshold of 3DGS, yields the lowest clone numbers, whereas GDAGS suppresses directionally inconsistent clones through gradient alignment constraints, resulting in intermediate cloning activity. GDAGS further ensures localized replenishment of directionally consistent clones while mitigating redundant accumulation of geometrically conflicting clones. For splitting operations, AbsGS generates excessive splits by indiscriminately utilizing gradient magnitudes, whereas 3DGS underperforms due to gradient direction conflicts. GDAGS effectively filters splittable Gaussians through adaptive criteria. Regarding total Gaussian counts, GDAGS exhibits a more stable trajectory and faster convergence compared to 3DGS and AbsGS, validating its optimized densification strategy in balancing geometric fidelity and computational efficiency.

Table 3: Generalization analysis on three datasets. SSIM↑ and PSNR↑ are higher-the-better; LPIPS↓ is lower-the-better. The best score is red.

| Datasets | Mip-NeRF360 | | | Tanks&Temples | | | Deep Blending | | |
|---|---|---|---|---|---|---|---|---|---|
| Methods | SSIM↑ | PSNR↑ | LPIPS↓ | SSIM↑ | PSNR↑ | LPIPS↓ | SSIM↑ | PSNR↑ | LPIPS↓ |
| MCMC-3DGS | 0.900 | 29.89 | 0.190 | 0.860 | 24.29 | 0.190 | 0.890 | 29.67 | 0.320 |
| MCMC-GDAGS | 0.900 | 29.80 | 0.150 | 0.860 | 24.19 | 0.150 | 0.910 | 29.59 | 0.240 |
| Compact-3DGS | 0.798 | 27.08 | 0.247 | 0.831 | 23.32 | 0.201 | 0.901 | 29.79 | 0.258 |
| Compact-GDAGS | 0.804 | 27.02 | 0.227 | 0.834 | 23.23 | 0.191 | 0.902 | 29.74 | 0.250 |

### 4.3.4 GENERALIZABILITY ANALYSIS

To demonstrate the generalizability of the proposed densification control strategy, we generalize our GDAGS module to two other advanced 3DGS frameworks: MCMC-3DGS (Kheradmand et al., 2024) and Compact-3DGS (Lee et al., 2024). As quantitatively shown in Table 3, the integration of GDAGS consistently improves the performance of both baselines, yielding superior results in terms of SSIM

and LPIPS. This demonstrates the broad applicability and effectiveness of our gradient-direction-aware densification control in enhancing reconstruction quality across different 3DGS architectures. As shown in the qualitative results of Figure 6, the model integrated with GDAGS performs better on object surfaces and geometric edges, such as room floors, wardrobe gaps, etc.

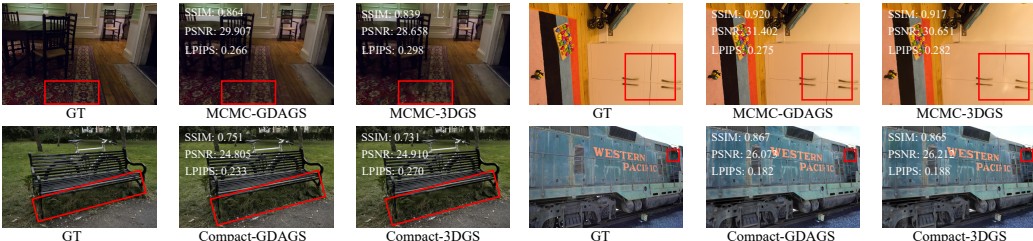

Figure 6: Qualitative analysis of GDAGS integrated in MCMC-3DGS and Compact-3DGS.

Table 4: Efficiency analysis of GDAGS and baseline models. TT(s) in the table represents training time (in seconds). The best score is red.

| Datasets | Mip-NeRF360 | | Tanks&Temples | | Deep Blending | |
|---|---|---|---|---|---|---|
| Methods | Training time(s)↓ | FPS↑ | Training time(s)↓ | FPS↑ | Training time(s)↓ | FPS↑ |
| 3DGS | 1594 | 134 | 912 | 154 | 1344 | 137 |
| Pixel-GS | 1811 | 89 | 1450 | 92 | 1640 | 90 |
| AbsGS | 1571 | 111 | 621 | 125 | 937 | 142 |
| Mini-splatting | 1320 | 386 | 896 | 432 | 1095 | 416 |
| GDAGS | 1140 | 188 | 555 | 220 | 898 | 196 |

## 4.4 EFFICIENCY ANALYSIS

We report the training time and inference time (FPS) of GDAGS compared to other baseline models in Table 4. As the results demonstrate, GDAGS achieves the fastest training convergence across all datasets. This efficiency stems from our gradient-direction-aware densification control, which strategically prioritizes the most impactful splitting and cloning operations. By avoiding redundant densification steps that contribute little to reconstruction quality, GDAGS not only produces a more compact scene representation but also reduces the total computational cost of training. Regarding the computational overhead of GCR, we argue it is negligible for two key reasons. First, the calculation of the GCR is computationally lightweight, involving only simple vector norms and summations on pre-existing gradients. Second, and more importantly, the GCR-driven densification control leads to a far more compact scene representation, often reducing the final number of Gaussians by hundreds of thousands to millions compared to 3DGS. This reduction directly accelerates subsequent optimization steps.

## 5 CONCLUSION

This paper proposes GDAGS, a dynamic adaptive Gaussian splatting framework that addresses critical challenges in 3D scene reconstruction, including over-densification, over-reconstruction, and memory inefficiency. By introducing the directional consistency metric (GCR) and nonlinear dynamic weighting, GDAGS enables direction-aware densification control: GCR discriminates between directionally consistent/divergent Gaussians, while dynamic weights independently regulate densification at the per-Gaussian level. This dual mechanism effectively suppresses redundant Gaussian proliferation, eliminates outlier distributions, and drives compact spatial representations without introducing additional heuristic densification thresholds.

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

## A  ALGORITHM

---

**Algorithm 1** Gradient-Direction-Aware Adaptive Densification

---

**Require:** Gaussian set $\mathcal{G}$, training views $\mathcal{V}$, initial thresholds $\tau$
**Ensure:** Optimized Gaussian set $\mathcal{G}'$
 1: Initialize gradient buffers $\{\nabla_{i,v}\} \leftarrow 0$, indicate the gradient of the $i$-th Gaussian at view angle $v$
 2: **for** each iteration $t = 1$ to $T$ **do**
 3:     **for** each view $v \in \mathcal{V}$ **do**
 4:         Render $\mathcal{G}$ via differentiable splatting
 5:         Compute 2D gradient map $\sum_{pixel} \nabla_{i,pixel}^v$ and $\sum_{pixel} \|\nabla_{i,p}^v\|_2$
 6:         Accumulate $\nabla_i^v$ for visible Gaussians
 7:     **end for**
 8:     **for** each Gaussian $g_i \in \mathcal{G}$ **do**
 9:         Compute $\mathcal{C}_i$ via Eq. equation 5
10:         Calculate $w_i$ using Eq. equation 6
11:     **end for**
12:     Compute $\nabla_{split}, \nabla_{clone}$
13:     Split if $\nabla_{split} > \tau_p$ & $\Sigma_{3D}^i > \tau_s$
14:     Clone if $\nabla_{clone} > \tau_p$ & $\Sigma_{3D}^i \leq \tau_s$
15: **end for**

---

## B  WEIGHT FUNCTION RESPONSE CURVE ANALYSIS

As shown in Figure 7, we systematically analyzed the influence of parameters $p$ and $\beta$ in the weight function. When $\beta = 25$ is fixed, increasing the value of $p$ will cause the weight curve to be in $\mathcal{C}_i$. When the value is smaller, it becomes steeper, which means stronger punishment is imposed on Gaussian beams with inconsistent directions. When $p = 15$ is fixed, $\beta$ mainly controls the overall magnitude of the weight function. These observation results are consistent with the quantitative analysis in the main text, verifying the rationality of parameter design.

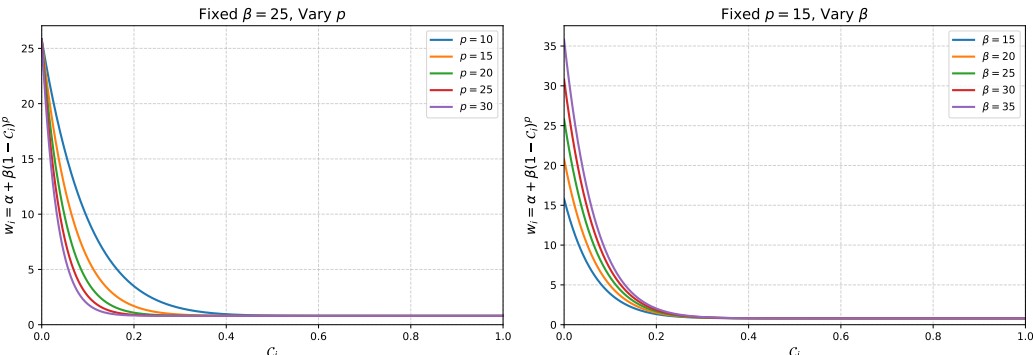

Figure 7: Corresponding curves of weights for different hyperparameters $p$ and $\beta$.

## C  ANALYZE

### C.1  HYPERPARAMETER SENSITIVITY ANALYSIS

The hyperparameters beta were set to 15, 20, 25, 30, and 35 for sensitivity analysis, and the results are shown in Figure 8. The results show that the setting of beta has a relatively small impact on overall performance, and increasing beta can slightly improve performance and slightly increase the required storage space.

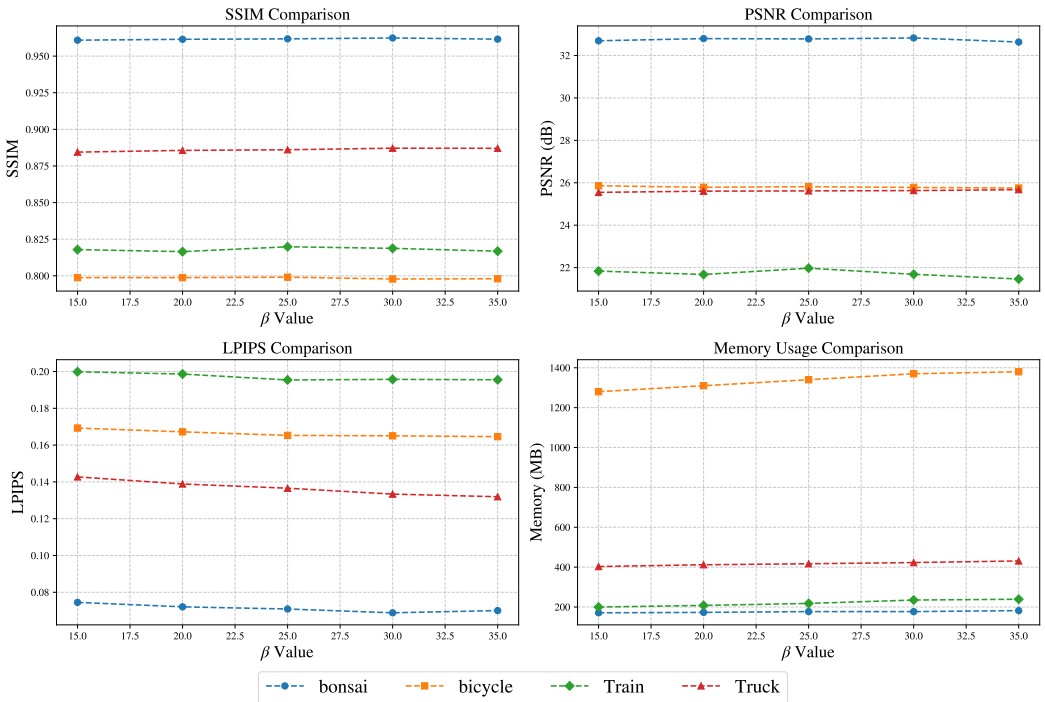

Figure 8: Performance of different hyperparameters $\beta$ in multiple datasets.

## C.2    DENSIFICATION PROCESS ANALYSIS

Figure 9 quantifies the Gaussian split/clone counts per densification round and the total Gaussian population dynamics during training. The results reveal distinct behavioral patterns across methods: In cloning, all three methods show consistent trends. 3DGS produces the most clones under its standard threshold, while AbsGS yields the fewest due to its doubled threshold. GDAGS suppresses directionally inconsistent clones via gradient alignment, resulting in intermediate numbers and ensuring localized replenishment without redundant accumulation. For splitting, AbsGS over-splits by using gradient magnitudes indiscriminately, whereas 3DGS under-splits due to gradient conflicts. GDAGS effectively filters splittable Gaussians with adaptive criteria. Overall, GDAGS exhibits a more stable trajectory and faster convergence in total Gaussian counts than 3DGS and AbsGS, confirming its balanced densification strategy between geometric fidelity and efficiency.

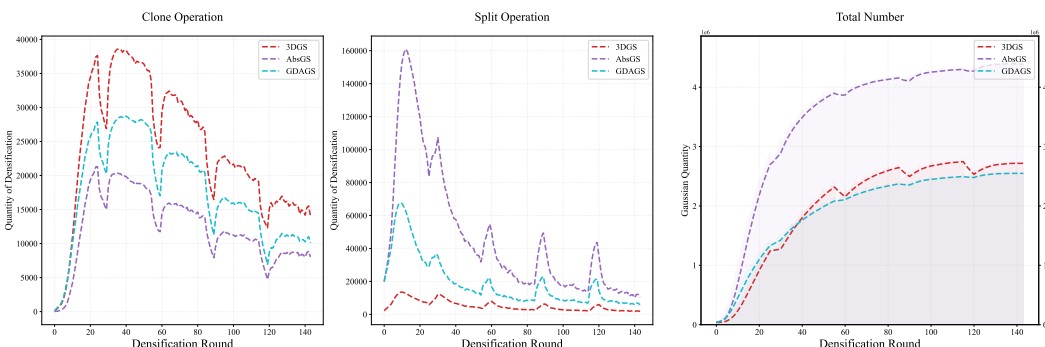

Figure 9: Visualization of different densification methods during the training process in the flowers sense.

Table 5: Comparison of training time between power function and exponential function. The best result is **bold**.

| Methods | Mip-NeRF360 | Tanks&Temples | Deep Blending |
|---|---|---|---|
| GDAGS | **1140s** | **555s** | **898s** |
| GDAGS-E | 1200s (+5.2%) | 612s (+10.3%) | 976s (+8.7%) |

Table 6: Comparison of performance between power function and exponential function. The best result is **bold**.

| Datasets | Mip-NeRF360 | | | Tanks&Temples | | | Deep Blending | | |
|---|---|---|---|---|---|---|---|---|---|
| Methods | SSIM↑ | PSNR↑ | LPIPS↓ | SSIM↑ | PSNR↑ | LPIPS↓ | SSIM↑ | PSNR↑ | LPIPS↓ |
| **GDAGS** | **0.839** | **28.02** | **0.145** | **0.852** | **23.79** | **0.165** | **0.905** | **29.70** | **0.235** |
| GDAGS-E | 0.837 | 27.96 | 0.146 | 0.851 | 23.62 | **0.165** | 0.903 | 29.60 | 0.237 |

## C.3 Nonlinear Dynamic Weighted Functions Analysis

Nonlinear functions are usually represented as power functions ($(1 - \mathcal{C}_i)^p$) and exponential functions ($e^{-p\mathcal{C}_i}$). GDAGS chooses to use the power function form for two main considerations. Firstly, from the perspective of computational load, the calculation of power functions is very efficient, while the calculation of exponential functions is more complex. The running time of torch.exp in PyTorch environment is longer than torch.pow. We construct a model consisting of exponential functions and named it GDAGS-E. We conduct additional exponential function ablation experiments, the training time and performance are reported in the Table 5 and Table 6. The results show that using exponential functions increases training time by 5%-10% compared to power functions.

Then, we analyze the properties of exponential and power functions themselves. We consider two candidate forms for the nonlinear weighting component, power function $f_p(x) = (1 - x)^p$ and exponential function $f_e(x) = e^{-px}$. To compare their sensitivity characteristics, we analyze the absolute values of their derivatives: $|f_p'(x)| = p(1 - x)^{p-1}$ and $|f_e'(x)| = pe^{-px}$. These derivatives represent the rate at which each weighting function responds to changes in gradient coherence. For a power function, it is a monotonically decreasing function over the domain $x \in [0, 1]$, with a value range of $f_p \in [0, 1]$. For a exponential function, it is a monotonically decreasing function over the domain $x \in [0, 1]$, with a value range of $f_p \in [pe^{-p}, 1]$. We let $f(x) = (1 - x)^p - e^{-px}$, whose derivative is $f' = p[e^{-px} - x^{p-1}]$, and the corresponding curve of the function is shown in the Figure 10. In the domain, $f(0) = 0$, $f(1) > 0$, the derivative is first less than 0 and then greater than 0, and the zero point $x = x_0$ keeps approaching 0. This indicates that the power function curve is always below the exponential function curve and as $p$ increases, the exponential function continues to approach the power function. When $0 < x < x_0$, $f' < 0$ indicates that the power function decreases faster than the exponential function in the $[0, x_0]$ interval. When $x_0 < x < 1$, $f' > 0$ indicates that the power function decreases more smoothly than the exponential function in the $[x_0, 1]$ interval. This is consistent with our design goal of selecting only Gaussians with significant gradient direction conflicts and maintaining consistent suppression for Gaussians with consistent gradient directions.

## C.4 Analysis of Abnormal Behavior of Different Models on Dataset Deep Blending

We observed that the performance of GDAGS on the Deep Blending dataset is slightly lower than that of Mini plating, and the number of Gaussians in Mini plating is significantly less than that of GDAGS. The performance metrics reconstructed on other datasets show the same trend as the Gaussian number, while the opposite trend is observed on the Deep Blending dataset. Based on this phenomenon, we propose a hypothesis that the relatively simple geometric shapes and textures of the Deep Blending dataset make them more prone to overfitting when modeling with excessive Gaussian numbers. To test this hypothesis, we introduced a Gaussian dropout mechanism into GDAGS during training, creating variants dubbed GDAGS-DROP (random dropout) and GDAGS-ODROP (opacity-weighted dropout). The results are reported in Table 7. The performance of GDAGS-DROP

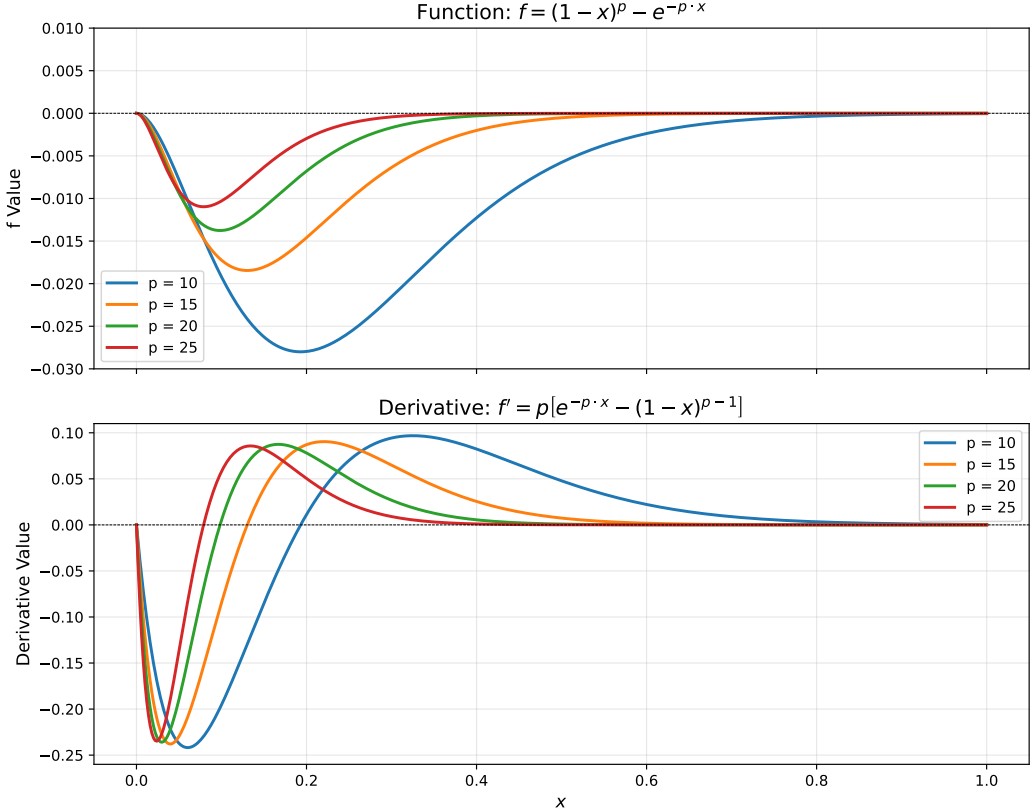

Figure 10: The function curve of the difference between power function and exponential function.

initially improves with a moderate dropout rate before declining, which confirms that overfitting is a relevant factor on this dataset. The optimally regularized variant, GDAGS-ODROP-5%, surpasses all Mini-splatting configurations in SSIM and PSNR. At the same time, we consulted some 3DGS related work, and one of the papers, ControlGS (Zhang et al., 2025), studied this situation. ControlGS proposes that as the number of Gaussians increases, scene reconstruction will go through four stages: underfitting, effective state, saturation, and overfitting. ControlGS also uses pruning to keep the model in an effective state. Notably, ControlGS exhibits a similar cross-dataset performance pattern: it underperforms against other baselines on complex datasets like Mip-NeRF-360 and Tanks&Temples but excels on Deep Blending, mirroring our findings.

Table 7: Performance comparison of GDAGS-DROP with Mini-splating-P (partial pruning) and Mini-splatting-D (only perform densification without pruning) on the Deep Blending dataset.

| Methods | Gaussians (M) | SSIM | PSNR | LPIPS |
|---|---|---|---|---|
| Mini-splatting | 0.35 | 0.908 | 29.98 | 0.253 |
| Mini-splatting-P | 1.55 | 0.906 | 29.92 | 0.230 |
| Mini-splatting-D | 4.63 | 0.906 | 29.88 | **0.211** |
| GDAGS (GDAGS-DROP-0%) | 1.71 | 0.905 | 29.70 | 0.235 |
| GDAGS-ODROP-5% | 0.74 | **0.909** | **30.03** | 0.241 |
| GDAGS-DROP-5% | 0.74 | 0.908 | 29.91 | 0.242 |
| GDAGS-DROP-10% | 0.58 | 0.908 | 29.88 | 0.247 |
| GDAGS-DROP-20% | **0.32** | 0.906 | 29.78 | 0.255 |

## C.5 GRADIENT SPARSE SCENE ANALYSIS

To analyze the behavior of GDAGS and baseline models in sparse gradient regions, we trained GDAGS-H, 3DGS-H, AbsGS-H, and Pixel-GS-H models respectively, where - H represents using 50% of the training views to increase the range of sparse gradient regions. As shown in Figure 11, GDAGS has fewer artifacts and better geometric edges in gradient sparse areas such as the sky, walls, and ground. Although the assumption of GDAGS is that Gaussian needs to receive sufficient gradient information, GDAGS still works well in sparse gradient regions.

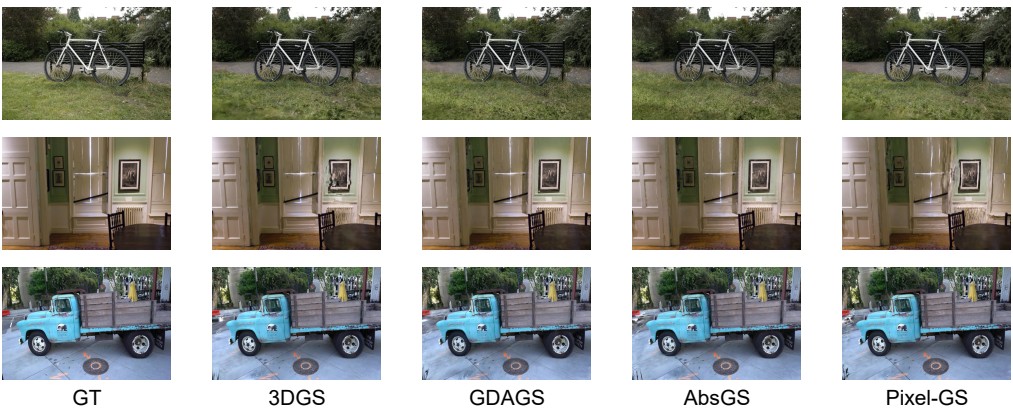

|    GT    |    3DGS    |    GDAGS    |    AbsGS    |    Pixel-GS    |

Figure 11: Qualitative results of GDAGS and baseline models in sparse views.

## C.6 ANALYSIS AND GUIDANCE OF HYPERPARAMETERS

First, we emphasize that all experimental results are obtained using a single, fixed set of hyperparameters ($\alpha = 0.8$, $\beta = 25$, $p = 15$) across all scenes and datasets, without any scene-specific tuning. The consistent improvements in rendering quality (PSNR, SSIM, LPIPS) and memory efficiency demonstrate the strong generalization capability and robustness of GDAGS under a fixed configuration.

Second, the roles of $\alpha$, $\beta$, and $p$ are well-defined and interpretable:

1. When the GCR approaches 1, the weight converges to $\alpha$. Since Gaussians with high GCR typically exhibit large gradient magnitudes, $\alpha$ (set between 0 and 1) acts as a suppression factor to prevent over-densification in well-reconstructed regions.

2. When GCR approaches 0, the weight becomes $\alpha + \beta$. These Gaussians often have small gradient norms due to directional conflicts. To promote their densification, we set $\beta$ to match the max_screen_size value from the original 3DGS implementation, ensuring compatibility and intuitive scaling.

3. The exponent $p$ controls the number of Gaussians selected for densification. Its effect is predictable: increasing $p$ restricts densification to fewer Gaussians, reducing memory usage at a potential cost to reconstruction quality; decreasing $p$ has the opposite effect. In practice, $\alpha$ and $\beta$ can remain unchanged with minimal impact on performance, and the value of $p$ can be adjusted according to different requirements (high rendering accuracy or low memory consumption).

Furthermore, our sensitivity analysis confirms that GDAGS maintains stable performance across a wide range of $p$ (10–30) and $\beta$ (15–35) values, reinforcing its generalizability without demanding extensive hyperparameter search.

# D    DETAILED RESULTS

The detailed performance metrics across a total of 13 different scenes from the three datasets are shown in Tables 8, 9, 10, 12, 13, and 14 while efficiency metrics are presented in Tables 11 and 15. Render quality comparisons are demonstrated in Figures 12 and 13, where significant differences are highlighted with red boxes and enlarged in the bottom-right corners. Figure 14 compares over-densification phenomena, with FPS values and the number of Gaussians annotated in the top-left corners.

Table 8: Per-scene quantitative results (SSIM) from the Mip-NeRF360.

|  | bicycle | flowers | garden | stump | treehill | room | counter | kitchen | bonsai |
|---|---|---|---|---|---|---|---|---|---|
| Plenoxels | 0.496 | 0.431 | 0.606 | 0.523 | 0.509 | 0.841 | 0.759 | 0.648 | 0.814 |
| INGP-Base | 0.491 | 0.450 | 0.649 | 0.574 | 0.518 | 0.855 | 0.798 | 0.818 | 0.890 |
| INGP-Big | 0.512 | 0.486 | 0.701 | 0.594 | 0.542 | 0.871 | 0.817 | 0.858 | 0.906 |
| Mip-NeRF360 | 0.685 | 0.583 | 0.813 | 0.744 | 0.632 | 0.913 | 0.894 | 0.920 | 0.941 |
| 3D-GS | 0.771 | 0.605 | 0.868 | 0.775 | 0.638 | 0.914 | 0.905 | 0.922 | 0.938 |
| AbsGS-0004 | 0.782 | 0.613 | 0.870 | 0.784 | 0.626 | 0.920 | 0.908 | 0.929 | 0.944 |
| Pixel-GS | 0.766 | 0.627 | 0.862 | 0.784 | 0.638 | 0.929 | 0.921 | 0.936 | 0.951 |
| GDAGS | **0.798** | **0.641** | **0.879** | **0.796** | **0.638** | **0.951** | **0.928** | **0.954** | **0.961** |

Table 9: Per-scene quantitative results (PSNR) from the Mip-NeRF360.

|  | bicycle | flowers | garden | stump | treehill | room | counter | kitchen | bonsai |
|---|---|---|---|---|---|---|---|---|---|
| Plenoxels | 21.912 | 20.097 | 23.495 | 20.661 | 22.248 | 27.594 | 23.624 | 23.420 | 24.669 |
| INGP-Base | 22.193 | 20.348 | 24.599 | 23.626 | 22.364 | 29.269 | 26.439 | 28.548 | 30.337 |
| INGP-Big | 22.171 | 20.652 | 25.069 | 23.466 | 22.373 | 29.690 | 26.691 | 29.479 | 30.685 |
| Mip-NeRF360 | 24.37 | **21.73** | 26.98 | 26.40 | **22.87** | 31.63 | **29.55** | 32.23 | **33.46** |
| 3D-GS | 25.246 | 21.520 | 27.410 | 26.550 | 22.490 | 30.632 | 28.700 | 30.317 | 31.980 |
| AbsGS-0004 | 25.373 | 21.298 | 27.579 | 26.766 | 22.074 | 31.582 | 28.968 | 31.774 | 32.283 |
| Pixel-GS | 25.263 | 21.677 | 27.393 | 26.938 | 22.343 | 31.952 | 29.270 | 31.950 | 32.687 |
| GDAGS | **25.738** | 21.711 | **27.839** | **27.064** | 22.217 | **32.566** | 29.531 | **32.698** | 32.839 |

Table 10: Per-scene quantitative results (LPIPS) from the Mip-NeRF360.

|  | bicycle | flowers | garden | stump | treehill | room | counter | kitchen | bonsai |
|---|---|---|---|---|---|---|---|---|---|
| Plenoxels | 0.506 | 0.521 | 0.386 | 0.503 | 0.540 | 0.4186 | 0.441 | 0.447 | 0.398 |
| INGP-Base | 0.487 | 0.481 | 0.312 | 0.450 | 0.489 | 0.301 | 0.342 | 0.254 | 0.227 |
| INGP-Big | 0.446 | 0.441 | 0.257 | 0.421 | 0.450 | 0.261 | 0.306 | 0.195 | 0.205 |
| Mip-NeRF360 | 0.301 | 0.344 | 0.170 | 0.261 | 0.339 | 0.211 | 0.204 | 0.127 | 0.176 |
| 3D-GS | 0.205 | 0.336 | 0.103 | 0.210 | 0.317 | 0.220 | 0.204 | 0.129 | 0.205 |
| AbsGS-0004 | 0.186 | 0.295 | 0.104 | 0.202 | 0.297 | 0.216 | 0.198 | 0.124 | 0.194 |
| Pixel-GS | 0.206 | 0.288 | 0.111 | 0.208 | 0.301 | 0.188 | 0.166 | 0.109 | 0.167 |
| GDAGS | **0.165** | **0.266** | **0.094** | **0.185** | **0.274** | **0.092** | **0.098** | **0.059** | **0.070** |

Table 11: Per-scene memory consumption (MB) from the Mip-NeRF360.

|  | bicycle | flowers | garden | stump | treehill | room | counter | kitchen | bonsai |
|---|---|---|---|---|---|---|---|---|---|
| 3D-GS* | 1508 | 813 | 1073 | 1100 | 1018 | 397 | 257 | 412 | 259 |
| Pixel-GS | 2040 | 1660 | 1980 | 1480 | 1690 | 584 | 592 | 721 | 487 |
| AbsGS-0004 | 1156 | 751 | 932 | 1021 | 962 | 234 | 192 | 249 | 234 |
| GDAGS | **1030** | **635** | **841** | **683** | **744** | **223** | **146** | **197** | **144** |

Table 12: Per-scene quantitative results (SSIM) from the Tanks & Temples and Deep Blending.

|  | Truck | Train | Dr Johnson | Playroom |
|---|---|---|---|---|
| Plenoxels | 0.774 | 0.663 | 0.787 | 0.802 |
| INGP-Base | 0.779 | 0.666 | 0.839 | 0.754 |
| INGP-Big | 0.800 | 0.689 | 0.854 | 0.779 |
| Mip-NeRF360 | 0.857 | 0.660 | **0.901** | 0.900 |
| 3D-GS | 0.879 | 0.802 | 0.899 | 0.906 |
| AbsGS-0004 | **0.886** | 0.820 | 0.900 | 0.907 |
| Pixel-GS | **0.886** | **0.826** | 0.887 | 0.905 |
| GDAGS | **0.886** | 0.819 | **0.901** | **0.909** |

Table 13: Per-scene quantitative results (PSNR) from the Tanks & Temples and Deep Blending.

|  | Truck | Train | Dr Johnson | Playroom |
|---|---|---|---|---|
| Plenoxels | 23.221 | 23.221 | 23.142 | 22.980 |
| INGP-Base | 23.260 | 20.170 | 27.750 | 19.483 |
| INGP-Big | 23.383 | 20.456 | 28.257 | 21.665 |
| Mip-NeRF360 | 24.912 | 19.523 | 29.140 | 29.657 |
| 3D-GS | 25.187 | 21.097 | 28.766 | 30.044 |
| AbsGS-0004 | **25.702** | 22.010 | 28.930 | 29.967 |
| Pixel-GS | 25.467 | **22.028** | 28.128 | 29.700 |
| GDAGS | 25.617 | 21.971 | **29.232** | **30.166** |

Table 14: Per-scene quantitative results (LPIPS) from the Tanks & Temples and Deep Blending.

|  | Truck | Train | Dr Johnson | Playroom |
|---|---|---|---|---|
| Plenoxels | 0.335 | 0.422 | 0.521 | 0.499 |
| INGP-Base | 0.274 | 0.386 | 0.381 | 0.465 |
| INGP-Big | 0.249 | 0.360 | 0.352 | 0.428 |
| Mip-NeRF360 | 0.159 | 0.354 | 0.237 | 0.252 |
| 3D-GS | 0.148 | 0.218 | 0.244 | 0.241 |
| AbsGS-0004 | 0.131 | 0.193 | 0.240 | **0.232** |
| Pixel-GS | **0.120** | **0.178** | 0.255 | 0.241 |
| GDAGS | 0.136 | 0.195 | **0.238** | 0.233 |

Table 15: Per-scene memory consumption (MB) from the Tanks & Temples and Deep Blending.

|  | Truck | Train | Dr Johnson | Playroom |
|---|---|---|---|---|
| 3D-GS* | 530 | 218 | 744 | 504 |
| AbsGS-0004 | 419 | 210 | 457 | 316 |
| Pixel-GS | 1200 | 904 | 1280 | 881 |
| GDAGS | **277** | **175** | **416** | **280** |

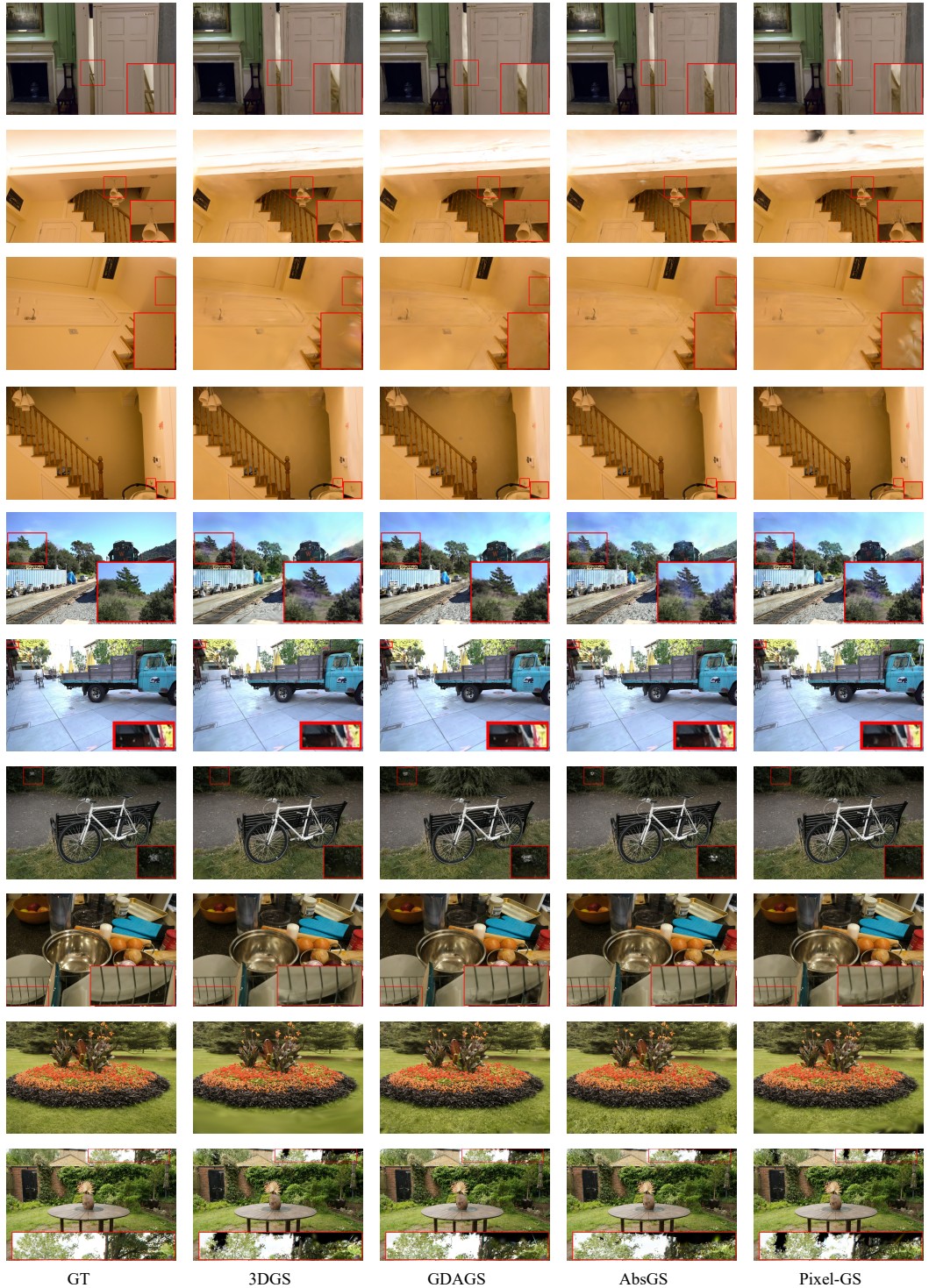

|  GT  |  3DGS  |  GDAGS  |  AbsGS  |  Pixel-GS  |

Figure 12: Supplementary qualitative analysis results.

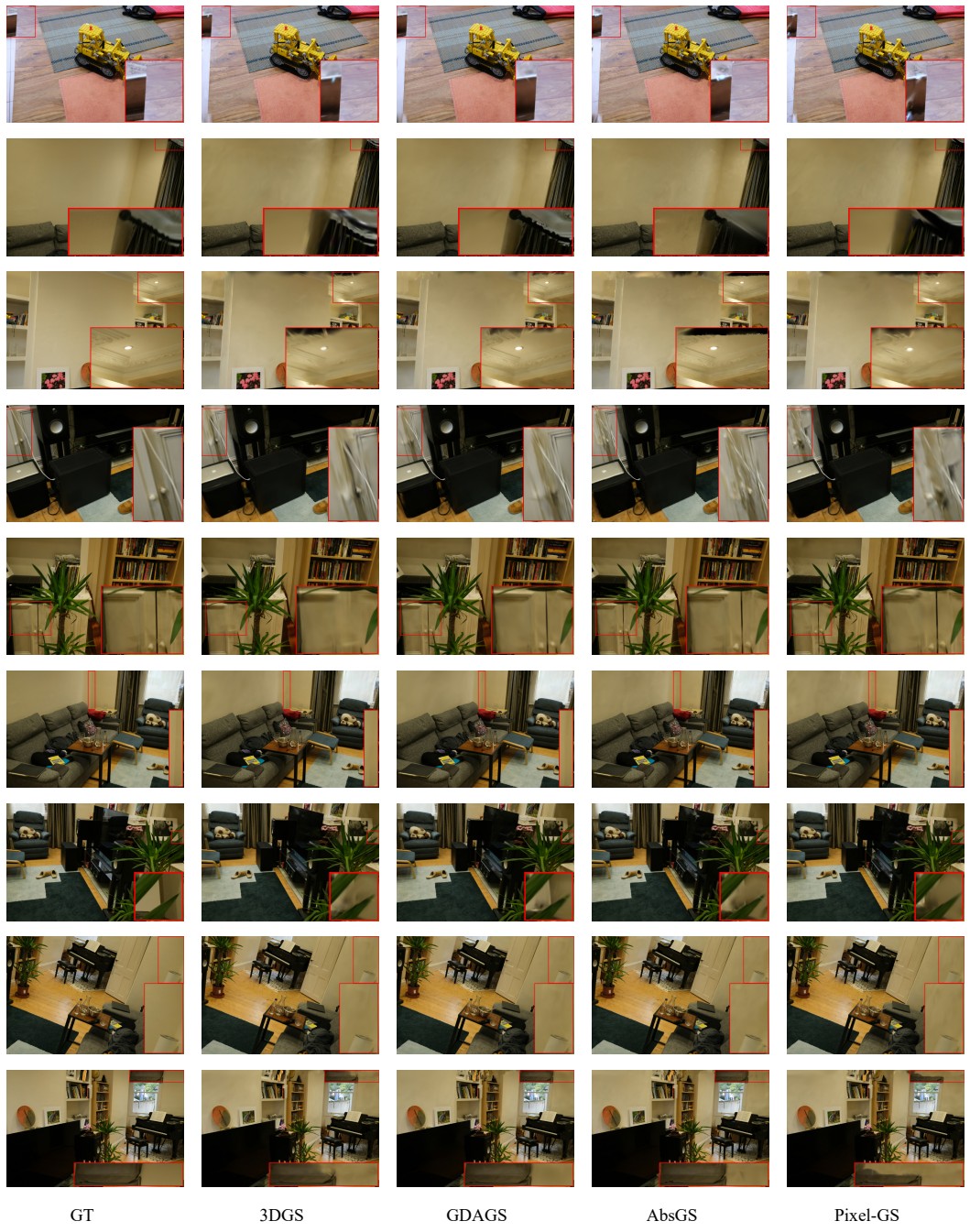

GT       3DGS       GDAGS       AbsGS       Pixel-GS

Figure 13: Supplementary qualitative analysis results.

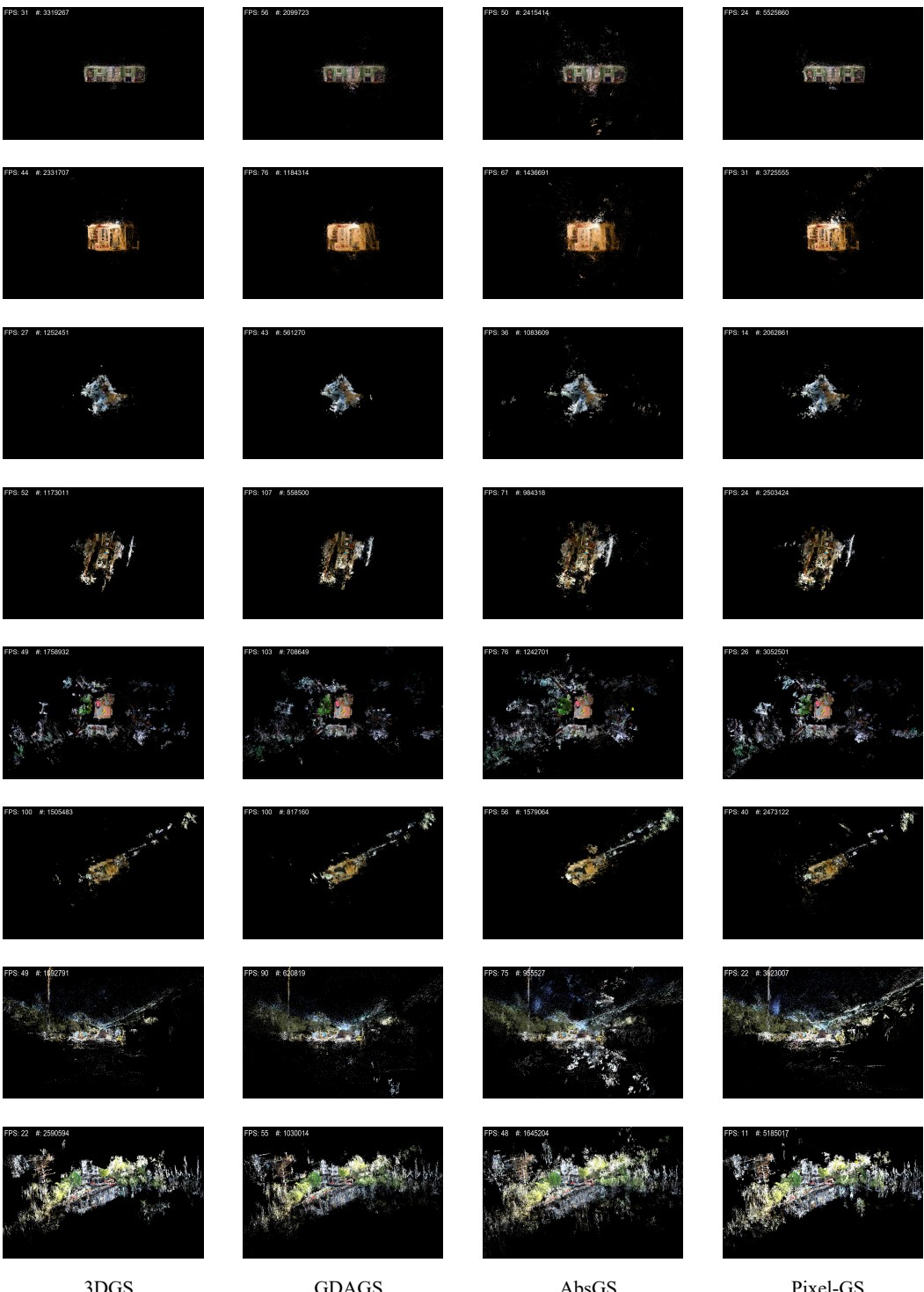

3DGS        GDAGS        AbsGS        Pixel-GS

Figure 14: Comparing over-densification phenomena.

## E    LIMITATIONS OF GDAGS

While GDAGS demonstrates significant improvements in adaptive densification control and memory efficiency, several limitations remain:

- Fixed Hyperparameters for Weighting Schemes: The hyperparameter $p$, which balances performance and efficiency, is fixed at 15 by default. While this value works well empirically, its optimal setting may vary across datasets or hardware configurations, requiring manual tuning for specific use cases.
- Challenges in extreme sparsity: In highly sparse regions with minimal gradient activity, GCR may struggle to distinguish between true outliers and under-reconstructed areas, leading to potential under-densification or over-suppression of Gaussians.

## F  LLM USAGE

Large language models (LLMs) were used to assist in the writing and polishing of this manuscript. Specifically, an LLM was employed to help refine language expression, improve readability, and enhance clarity in certain sections of the paper. Tasks included sentence rephrasing, grammar checking, and improving the overall flow of the text. It is important to note that the LLM was not involved in the conception of ideas, research methodology, experimental design, or data analysis. All scientific content, conceptual development, and analytical results were generated solely by the authors. The use of the LLM was strictly limited to improving the linguistic quality of the manuscript and did not extend to any scientific reasoning or interpretation. The authors take full responsibility for the entire content of the manuscript, including any text modified or suggested by the LLM. We have carefully reviewed all AI-assisted content to ensure it adheres to ethical guidelines and accurately reflects the original intent and rigor of our research.

