# OpenReview forum: "Gradient-Direction-Aware Density Control for 3D Gaussian Splatting"
_ICLR.cc/2026/Conference — ICLR 2026 Poster_

### Official Review · Reviewer_52Pb · 2025-10-30

**Soundness:** 3
**Presentation:** 3
**Contribution:** 2
**Rating:** 6
**Confidence:** 3

**Summary:**

This paper addresses a fundamental flaw in the "densification" process of 3D Gaussian Splatting (3DGS)—the mechanism that decides where to add more detail. The original 3DGS method relies solely on the magnitude (or norm) of the position gradient (which can be seen as the "rendering error signal") while completely ignoring its direction. To resolve this fundamental issue, the paper proposes a Gradient-Direction-Aware density control framework (GDAGS). Its core idea is to quantify and leverage gradient directional information to make smarter decisions. The method consists of two main components: The Gradient Coherence Ratio (GCR) and The Nonlinear Dynamic Weighting System. The method not only resolves the blurring issue and improves rendering quality but, more importantly, it reduces redundancy, producing compact 3D models than AbsGS.

**Strengths:**

1. The paper is well-written and easy to follow.
2. The work's originality lies in its more complete diagnosis of a known issue. While prior work like AbsGS had already identified the problem of gradient cancellation causing blur, this paper correctly points out that this is only half of the story; gradient amplification in aligned regions is an equally important problem that leads to model bloat. The proposed Gradient Coherence Ratio (GCR) is a novel and intuitive metric to directly measure this directional consistency. Using this metric to create a dual-purpose control system—one that simultaneously encourages splits in chaotic regions and suppresses them in stable ones—is a clever and well-motivated approach.

**Weaknesses:**

1. The authors correctly identify in their limitations section that the GCR metric may be unreliable in very sparse regions with little gradient information. However, this critical failure mode is only mentioned briefly and not explored empirically.
2. The method introduces new hyperparameters for its weighting function. Although a sensitivity analysis is provided in the appendix, the paper offers little intuition or practical guidance on how these should be set. This lack of guidance could make the method difficult for others to apply effectively to new and different scenes.

**Questions:**

1. To better understand the method's limitations, the authors should include a targeted experiment on a scene with known sparse viewpoints or textureless surfaces. A qualitative analysis showing how GDAGS behaves in these low-information areas compared to the baseline would be very instructive. Does it fail gracefully by simply not adding primitives, or does it make poor decisions? A discussion of potential fallback strategies in such cases would also strengthen the work.
2. The authors should report end-to-end training times and compare them against the key baselines. This is essential for a fair assessment of the trade-offs. It would also be helpful to include a brief profiling analysis that quantifies the specific overhead of the GCR computation step.

---

> ### Author Response · Authors · 2025-11-19
>
> We sincerely thank Reviewer 52Pb for their thorough review and valuable feedback. We have carefully addressed each of the concerns raised, as detailed below.
>
> **W1 & Q1**: The authors correctly identify in their limitations section that the GCR metric may be unreliable in very sparse regions with little gradient information. However, this critical failure mode is only mentioned briefly and not explored empirically...Does it fail gracefully by simply not adding primitives, or does it make poor decisions? A discussion of potential fallback strategies in such cases would also strengthen the work.
>
> **A**: We thank the reviewer for highlighting this important limitation. We randomly reduced the training views by half on the existing dataset to simulate sparse view scenarios and compared GDAGS with the baseline model. The results are now included in Appendix D. Our findings indicate that in such low-information regions, GDAGS tends to under-densify due to the lack of reliable gradient signals, but it does not introduce severe artifacts or outliers. This behavior is preferable to over-densification, which can lead to memory bloat and noisy renderings. One feasible backup solution we consider is to supplement the sparse gradient information with additional information constrained by multi view consistency.
>
> **W2**: The method introduces new hyperparameters for its weighting function. Although a sensitivity analysis is provided in the appendix, the paper offers little intuition or practical guidance on how these should be set. This lack of guidance could make the method difficult for others to apply effectively to new and different scenes.
>
> **A**: As we respond to reviewer dpMa, each hyperparameter has a fixed meaning and strong generalization. In practice, the only parameter that needs to be adjusted according to requirements is $p$.
>
> We address this concern from two perspectives. First, we emphasize that all experimental results reported in the main manuscript were obtained using a single, fixed set of hyperparameters ($\alpha = 0.8$,  $\beta=25$, $p = 15$) across all scenes and datasets, without any scene-specific tuning. The consistent improvements in rendering quality (PSNR, SSIM, LPIPS) and memory efficiency demonstrate the strong generalization capability and robustness of GDAGS under a fixed configuration.
>
> Second, the roles of $\alpha$, $\beta$, and $p$ are well-defined and interpretable:
>
> 1. When the GCR approaches 1, the weight converges to $\alpha$. Since Gaussians with high GCR typically exhibit large gradient magnitudes, $\alpha$ (set between 0 and 1) acts as a suppression factor to prevent over-densification in well-reconstructed regions.
> 2. When GCR approaches 0, the weight becomes $\alpha+\beta$. These Gaussians often have small gradient norms due to directional conflicts. To promote their densification, we set $\beta$ to match the max_screen_size value from the original 3DGS implementation, ensuring compatibility and intuitive scaling.
> 3. The exponent $p$ controls the number of Gaussians selected for densification. Its effect is predictable: increasing $p$ restricts densification to fewer Gaussians, reducing memory usage at a potential cost to reconstruction quality; decreasing $p$ has the opposite effect. In practice, $\alpha$ and $\beta$ can remain unchanged with minimal impact on performance, and the value of $p$ can be adjusted according to different requirements (high rendering accuracy or low memory consumption).
>
> Furthermore, our sensitivity analysis (Section 4.3.2 and Appendix D) confirms that GDAGS maintains stable performance across a wide range of $p$ (10–30) and $\beta$ (15–35) values, reinforcing its generalizability without demanding extensive hyperparameter search.

---

> > ### Author Response · Authors · 2025-11-19
> >
> > **Q2**: The authors should report end-to-end training times and compare them against the key baselines. This is essential for a fair assessment of the trade-offs. It would also be helpful to include a brief profiling analysis that quantifies the specific overhead of the GCR computation step.
> >
> > **A**: Thank you for this suggestion. We have now added comprehensive training time comparisons in the revised manuscript. The table below shows the total training time (in seconds) for each method across the three benchmark datasets:
> >
> > Table 1: Comparison of training time for different baseline models and GDAGS, as well as the proportion of GDAGS time in the densification period.
> >
> > | Methods                      | Mip-NeRF360 | Tanks&Temples | Deep Blending |
> > | :--------------------------- | :---------: | :-----------: | :-----------: |
> > | 3DGS                         |    1594     |      912      |     1344      |
> > | Pixel-GS                     |    1811     |     1450      |     1640      |
> > | AbsGS                        |    1571     |      621      |      937      |
> > | Taming 3DGS                  |    1920     |     1200      |     1320      |
> > | Mini-splatting               |    1320     |      896      |     1095      |
> > | GDAGS                        |  **1140**   |    **555**    |    **898**    |
> > | GDAGS (Densification Period) |  3.6 (3%)   |   2.5 (4%)    |    3.1(3%)    |
> >
> > As the results demonstrate, GDAGS achieves the fastest training convergence across all datasets. This efficiency stems from our gradient-direction-aware densification control, which strategically prioritizes the most impactful splitting and cloning operations. By avoiding redundant densification steps that contribute little to reconstruction quality (a key weakness of methods like AbsGS and Pixel-GS), GDAGS not only produces a more compact scene representation but also reduces the total computational cost of training. Regarding the computational overhead of GDAGS, we argue it is negligible for two key reasons. First, the calculation of the GCR is computationally lightweight, involving only simple vector norms and summations on pre-existing gradients. Second, and more importantly, the GCR-driven densification control leads to a far more compact scene representation, often reducing the final number of Gaussians by hundreds of thousands to millions compared to 3DGS (Figure 5). This reduction directly accelerates subsequent optimization steps. We have marked FPS and Gaussian total number # in Figure 12 of Appendix E, which strongly supports this point.

---

> > > ### Author Response · Authors · 2025-11-26
> > > **Looking forward to your reply**
> > >
> > > We sincerely appreciate your thoughtful suggestions on exploring limitations. We have conducted new experiments on sparse-view scenarios, added practical hyperparameter guidance, and included full efficiency analyses as requested. We hope you find the revised manuscript much strengthened.

---

### Official Review · Reviewer_5e9Z · 2025-11-01

**Soundness:** 3
**Presentation:** 3
**Contribution:** 2
**Rating:** 4
**Confidence:** 4

**Summary:**

This paper proposes GDAGS, a framework that addresses the over-densification, over-reconstruction, and memory inefficiency issues caused by the ill-posed densification mechanism in the original 3DGS, leveraging the directional consistency method and online dynamic weighting schemes. Experimental results show the efficacy of the proposed method.

**Strengths:**

1. The paper is well writen and easy to follow.
2. This paper targets a general and key component in the 3DGS pipeline and provides a simple yet effective solution.
3. The experimental results show promising performance improvements.

**Weaknesses:**

1. The proposed GDAGS introduces additional computational overhead during the optimization process. I believe it is necessary to report a comparison of training and testing times to better characterize the efficiency of the proposed method.

2. In Eq. (1), the variable \( i \) is not clearly defined. Does it refer to the Gaussian kernel? In addition, how is the number of views \( V \) determined in the experiments? Has the effect of different \( V \) values on performance been examined?

**Questions:**

Please see the weaknesses.

---

> ### Author Response · Authors · 2025-11-18
>
> **Q1**: The proposed GDAGS introduces additional computational overhead during the optimization process. I believe it is necessary to report a comparison of training and testing times to better characterize the efficiency of the proposed method.
>
> **A**: Thank you for this suggestion. We have now added comprehensive training time comparisons in the revised manuscript. The table below shows the total training time (in seconds) and testing time (FPS) for each method across the three benchmark datasets:
>
> Table 1: Comparison of training time between different baseline models and GDAGS.
>
> | Methods        | Mip-NeRF360 | Tanks&Temples | Deep Blending |
> | :------------- | :---------: | :-----------: | :-----------: |
> | 3DGS           |    1594     |      912      |     1344      |
> | Pixel-GS       |    1811     |     1450      |     1640      |
> | AbsGS          |    1571     |      621      |      937      |
> | Taming 3DGS    |    1920     |     1200      |     1320      |
> | Mini-splatting |    1320     |      896      |     1095      |
> | GDAGS          |  **1140**   |    **555**    |    **898**    |
>
> Table 2: Comparison of testing time between different baseline models and GDAGS.
>
> | Methods        | Mip-NeRF360 | Tanks&Temples | Deep Blending |
> | :------------- | :---------: | :-----------: | :-----------: |
> | 3DGS           |     134     |      154      |      137      |
> | Pixel-GS       |     89      |      92       |      90       |
> | AbsGS          |     111     |      125      |      142      |
> | Mini-splatting |   **386**   |    **432**    |    **416**    |
> | GDAGS          |     188     |      220      |      196      |
>
> As the results demonstrate, GDAGS achieves the fastest training convergence across all datasets. This efficiency stems from our gradient-direction-aware densification control, which strategically prioritizes the most impactful splitting and cloning operations. By avoiding redundant densification steps that contribute little to reconstruction quality (a key weakness of methods like AbsGS and Pixel-GS), GDAGS not only produces a more compact scene representation but also reduces the total computational cost of training. Regarding the computational overhead of GDAGS, we argue it is negligible for two key reasons. First, the calculation of the GCR is computationally lightweight, involving only simple vector norms and summations on pre-existing gradients. Second, and more importantly, the GCR-driven densification control leads to a far more compact scene representation, often reducing the final number of Gaussians by hundreds of thousands to millions compared to 3DGS (Figure 5). This reduction directly accelerates subsequent optimization steps. We have marked FPS and Gaussian total number # in Figure 12 of Appendix E, which strongly supports this point.
>
> **Q2**: In Eq. (1), the variable ( i ) is not clearly defined. Does it refer to the Gaussian kernel? In addition, how is the number of views ( V ) determined in the experiments? Has the effect of different ( V ) values on performance been examined?
>
> **A**: We apologize for the lack of clarity and thank the reviewer for pointing this out. The subscript *i* in the referenced equations and algorithm consistently refers to the *i*-th Gaussian primitive within the set, and we have clarified this definition in the revised manuscript. Regarding the number of views *V*, it denotes the count of training viewpoints from which a given Gaussian is visible during a densification step, in alignment with the practice of the original 3DGS. In our experiments, we utilized all available training views under which the Gaussian projects to a non-negligible size for computing its GCR and gradients, without imposing any artificial limit on *V*.

---

> > ### Author Response · Authors · 2025-11-26
> > **Looking forward to your reply**
> >
> > Thank you for your comments on clarity and efficiency. We clarified the variable definitions in the equation and reported a comprehensive comparison of training and inference time in the revised manuscript. We hope that these new additions can meet your expectations.

---

### Official Review · Reviewer_dpMa · 2025-11-01

**Soundness:** 3
**Presentation:** 2
**Contribution:** 3
**Rating:** 6
**Confidence:** 5

**Summary:**

3DGS introduced an adaptive density control algorithm to grow the number of Gaussians to fit the underlying scene structure. However, it is relatively suboptimal in properly splitting and cloning Gaussians, an issue that even prior work such as AbsGS could not fully address. This paper identifies that this issue arises from: 1) gradient cancellation due to diverging sub-gradient directions, and 2) exaggerated gradients caused by the simple aggregation of absolute values of diverging sub-gradients in local regions. To capture the directional consistency of sub-gradients, the authors define the Gradient Coherence Ratio (GCR; Equation 5) and modulate gradients using a nonlinear weighting function defined in Equation 6. Using this criterion, they not only promote the splitting of large Gaussians with diverging sub-gradients but also suppress the cloning of small Gaussians with diverging sub-gradients. Experimental results demonstrate state-of-the-art scene reconstruction quality while maintaining lower memory consumption on standard benchmarking datasets.

**Strengths:**

This paper properly identifies the shortcomings of the prior method (AbsGS) and achieves the best control of the number of Gaussians during training by addressing this problem.

**Weaknesses:**

Please see Questions section for my major concerns.

Presentation issues:
- Math error in Equation 3. The expansion of $T_k$ is incorrect. It must be $\prod_{j=1}^{k-1} (1-\alpha_jG'_j(\textbf{x}'))$.
- Typo in line 273: $(1 − C_i)^a$ → $(1 − C_i)^p$.
- Clarify the unit of x-axis in Figure 5. It seems k (thousand).

**Questions:**

* Please provide the training duration in the experiments. This will help readers understand the training efficiency of the method.
* In Figure 3, some images show that GDAGS fails to reconstruct particular areas compared to vanilla 3DGS, which weakens the authors’ argument that GDAGS avoids over-densification and over-reconstruction issues. Do the authors have an explanation for why GDAGS renders these artifacts? For example, GDAGS produces noisy artifacts on the crown molding (top-left of the first-row image) and blobby artifacts in the area between trees and the sky (top-left of the second-row image).
* One major limitation is that the newly introduced hyperparameters $\alpha, \beta, p$ are tuned specifically for different scenes. They need to be searched heuristically to find the best trade-off between quality and efficiency (VRAM usage), which is cumbersome. Is there a learnable approach for these parameters?
* In Figure 5, the authors show the dynamics of the number of Gaussians associated with clone/split operations. According to the graphs, AbsGS tends to split Gaussians far more frequently than GDAGS, yet AbsGS retains poorer reconstruction quality than GDAGS according to Table 1. This seems counter-intuitive because generally, propagating more Gaussians should improve the reconstruction of complex structures. What is the authors’ explanation for why AbsGS has lower PSNR despite splitting Gaussians much more than GDAGS? In other words, what is the key factor that allows GDAGS to achieve the highest rendering performance without splitting as many Gaussians?

---

> ### Author Response · Authors · 2025-11-18
>
> We sincerely thank Reviewer dpMa for their thorough review and valuable feedback. We have carefully addressed each of the concerns raised, as detailed below.
>
> **Presentation issues**: Incorrect formulas and chart issues.
>
> **Response**: Thank you to the reviewer for pointing out these issues. We have made comprehensive corrections in the revised manuscript. The x-axis in Figure 5 of our manuscript represents the densification iteration steps. According to the settings in the original 3DGS, the total number of iterations is 30000. The first 500 steps are used for preheating, and steps 500 to 15000 are used for densification. The interval between densification is 100 steps, so densification is performed a total of 145 times, and steps 15000 to 30000 are not performed.
>
> **Q1**: Please provide the training duration in the experiments. This will help readers understand the training efficiency of the method.
>
> **A**: Thank you for this suggestion. We have now added comprehensive training time comparisons in the revised manuscript. The table below shows the total training time (in seconds) for each method across the three benchmark datasets:
>
> Table 1: Comparison of training time between different baseline models and GDAGS.
>
> | Methods        | Mip-NeRF360 | Tanks&Temples | Deep Blending |
> | :------------- | :---------: | :-----------: | :-----------: |
> | 3DGS           |    1594     |      912      |     1344      |
> | Pixel-GS       |    1811     |     1450      |     1640      |
> | AbsGS          |    1571     |      621      |      937      |
> | Taming 3DGS    |    1920     |     1200      |     1320      |
> | Mini-splatting |    1320     |      896      |     1095      |
> | GDAGS          |  **1140**   |    **555**    |    **898**    |
>
> As the results demonstrate, GDAGS achieves the fastest training convergence across all datasets. This efficiency stems from our gradient-direction-aware densification control, which strategically prioritizes the most impactful splitting and cloning operations. By avoiding redundant densification steps that contribute little to reconstruction quality (a key weakness of methods like AbsGS and Pixel-GS), GDAGS not only produces a more compact scene representation but also reduces the total computational cost of training.
>
> **Q2**: In Figure 3, some images show that GDAGS fails to reconstruct particular areas compared to vanilla 3DGS, which weakens the authors’ argument that GDAGS avoids over-densification and over-reconstruction issues. Do the authors have an explanation for why GDAGS renders these artifacts? For example, GDAGS produces noisy artifacts on the crown molding (top-left of the first-row image) and blobby artifacts in the area between trees and the sky (top-left of the second-row image).
>
> **A**: We thank the reviewer for their keen observation. The mentioned artifacts primarily occur in regions such as distant sky or wall areas, where gradient signals can become relatively sparse, causing the GCR metric to be less reliable. In such cases, GDAGS may conservatively under-densify to avoid introducing redundant Gaussians, which can occasionally lead to minor reconstruction gaps or localized noise. However, it is important to note that these artifacts are relatively rare and localized, while GDAGS consistently reduces widespread over-reconstruction and over-densification across most of the scene (see Figures 10 and 11 in Appendix E). In fact, as shown in Figure 3, the quantitative metrics (SSIM, PSNR, and LPIPS) of GDAGS still outperform those of the vanilla 3DGS, demonstrating its overall effectiveness in balancing reconstruction quality and structural compactness.

---

> > ### Author Response · Authors · 2025-11-18
> >
> > **Q3**: One major limitation is that the newly introduced hyperparameters **α、β、p** are tuned specifically for different scenes. They need to be searched heuristically to find the best trade-off between quality and efficiency (VRAM usage), which is cumbersome. Is there a learnable approach for these parameters?
> >
> > **A**: This is an excellent point. As we respond to reviewer QXqY, each hyperparameter has a fixed meaning and strong generalization. In practice, the only parameter that needs to be adjusted according to requirements is $p$. We are trying to use a small MLP to automatically adjust these parameters.
> >
> > We address this concern from two perspectives. First, we emphasize that all experimental results reported in the main manuscript were obtained using a single, fixed set of hyperparameters ($\alpha = 0.8$,  $\beta=25$, $p = 15$) across all scenes and datasets, without any scene-specific tuning. The consistent improvements in rendering quality (PSNR, SSIM, LPIPS) and memory efficiency demonstrate the strong generalization capability and robustness of GDAGS under a fixed configuration.
> >
> > Second, the roles of $\alpha$, $\beta$, and $p$ are well-defined and interpretable:
> >
> > 1. When the GCR approaches 1, the weight converges to $\alpha$. Since Gaussians with high GCR typically exhibit large gradient magnitudes, $\alpha$ (set between 0 and 1) acts as a suppression factor to prevent over-densification in well-reconstructed regions.
> > 2. When GCR approaches 0, the weight becomes $\alpha+\beta$. These Gaussians often have small gradient norms due to directional conflicts. To promote their densification, we set $\beta$ to match the max_screen_size value from the original 3DGS implementation, ensuring compatibility and intuitive scaling.
> > 3. The exponent $p$ controls the number of Gaussians selected for densification. Its effect is predictable: increasing $p$ restricts densification to fewer Gaussians, reducing memory usage at a potential cost to reconstruction quality; decreasing $p$ has the opposite effect. In practice, $\alpha$ and $\beta$ can remain unchanged with minimal impact on performance, and the value of $p$ can be adjusted according to different requirements (high rendering accuracy or low memory consumption).
> >
> > Furthermore, our sensitivity analysis (Section 4.3.2 and Appendix D) confirms that GDAGS maintains stable performance across a wide range of $p$ (10–30) and $\beta$ (15–35) values, reinforcing its generalizability without demanding extensive hyperparameter search.
> >
> > **Q4**: In Figure 5, the authors show the dynamics of the number of Gaussians associated with clone/split operations. According to the graphs, AbsGS tends to split Gaussians far more frequently than GDAGS, yet AbsGS retains poorer reconstruction quality than GDAGS according to Table 1. This seems counter-intuitive because generally, propagating more Gaussians should improve the reconstruction of complex structures. What is the authors’ explanation for why AbsGS has lower PSNR despite splitting Gaussians much more than GDAGS? In other words, what is the key factor that allows GDAGS to achieve the highest rendering performance without splitting as many Gaussians?
> >
> > **A**: Thank you for this insightful question. The key difference lies in the quality and placement of new Gaussians, not just their quantity. AbsGS enforces uniform gradient direction, which amplifies gradient magnitudes indiscriminately. This leads to excessive and often unnecessary splitting, even in regions that are already well-reconstructed. Many of these new Gaussians are redundant or poorly positioned, contributing little to geometric accuracy and even introducing noise. Figure 12 in Appendix E strongly illustrates this situation, as it can be clearly observed in the top view of the point cloud that AbsGS creates a large number of Gaussian points outside the geometry of the object, which do not contribute to improving rendering performance.
> >
> > In contrast, GDAGS uses the GCR to detect directional conflicts, which are a strong indicator of regions where a single Gaussian is insufficient to represent the underlying geometry. By splitting only when directional inconsistency is high, GDAGS ensures that each new Gaussian is strategically placed to resolve genuine geometric ambiguity. This results in a more efficient and accurate Gaussian distribution, which explains why GDAGS achieves superior reconstruction quality (higher PSNR, lower LPIPS) with fewer Gaussians and less memory.

---

> ### Comment · Reviewer_dpMa · 2025-11-18
> **Additional question**
>
> Thank to the authors for detailed explanation. I have few trailing comments and questions.
>
> **Figure 5**:
> I initially thought the x-axis in Figure 5 indicated the training iteration, but I realized I was mistaken after reading your response. I think “densification round,” the term you used in line 429, is a better fit for the x-axis title than “iteration step.”
>
>
> **Q1**:
> The Table 1 that the authors newly provided clearly highlights the effectiveness of prioritizing impactful splitting/cloning using GCR. However, Table 2, which you provided to reviewer 5e9Z, shows a result slightly different from my expectations: the testing speed of Mini-splatting is nearly twice as fast as GDAGS. I think this may be because the authors directly imported the values from their paper, where the experimental setting involves extremely aggressive pruning of Gaussians. This leaves me wondering whether the low rendering quality metrics (SSIM/PSNR/LPIPS) of Mini-splatting in Table 1 of manuscript might be a consequence of such extreme pruning. To give readers better insight, could you check whether GDAGS still shows better quality when Mini-splatting’s pruning ratio is adjusted to maintain the same number of Gaussians as GDAGS?
>
> **Q2**:
> I understand the authors’ explanation regarding the artifacts in the figures. The GCR method is fundamentally defined based on pixel-wise subgradients, and it works properly only when the Gaussians receive sufficient gradient signals. This is an important assumption of GCR, and I suggest that clearly stating this assumption in the main text would improve the quality of your manuscript.
>
> **Q3**:
> I understand the authors’ explanation regarding method generalization. However, I noticed that the authors stated, “This dual mechanism effectively suppresses … *without relying on heuristic threshold tuning*” in Lines 484-485. Doesn’t this contradict what is actually done in the manuscript (particularly section 4.3.2 and Figure 4)? Although $\alpha$ and $\beta$ are fixed and do not require tuning, $p$ still needs to be selected to achieve the best trade-off between memory and rendering quality based on the user’s requirements.
>
> **Q4**:
> Thanks for the explanation. It is clear to me now.

---

> > ### Author Response · Authors · 2025-11-19
> >
> > Thank you for your positive feedback and these valuable follow-up questions. We address each point below to further improve the clarity and rigor of our work.
> >
> > **Figure 5**: I initially thought the x-axis in Figure 5 indicated the training iteration, but I realized I was mistaken after reading your response. I think “densification round,” the term you used in line 429, is a better fit for the x-axis title than “iteration step.”
> >
> > **A**: Thank you for this excellent suggestion. We agree that "Densification Round" is a more precise label for the x-axis. We have updated Figure 5 and the corresponding caption in the revised manuscript accordingly.
> >
> > **Q1**: **Comparison with Mini-splatting under matched Gaussian counts.**
> >
> > **A**: This is a very insightful question. You are right, the results of Mini-splating in our report were obtained directly from their paper, which may be the result of their aggressive pruning strategy, resulting in lower SSIM/PSNR/LPIPS scores in Table 1. In order to conduct a more fair comparison among peers, we conducted an additional experiment in which we adjusted the pruning intensity of Mini-splating to approximately match the Gaussian number of GDAGS. The results are reported in the table below. When the Gaussian counts are roughly the same, the quality indicators of Mini plating have improved, but still lag behind GDAGS.
> >
> > Table 1: Performance comparison of GDAGS with Mini-splating-P (partial pruning) and Mini-splatting-D (only perform densification without pruning) on the Mip-NeRF 360 dataset.
> >
> > | Methods          | \# Gaussians (M) | SSIM      | PSNR      | LPIPS     |
> > | :--------------- | :--------------- | :-------- | :-------- | --------- |
> > | Mini-splatting   | **0.49**         | 0.822     | 27.34     | 0.217     |
> > | Mini-splatting-P | 2.21             | 0.828     | 27.47     | 0.183     |
> > | Mini-splatting-D | 4.69             | 0.831     | 27.51     | 0.176     |
> > | GDAGS            | 2.39             | **0.839** | **28.02** | **0.145** |
> >
> > Table 2: Performance comparison of GDAGS with Mini-splating-P (partial pruning) and Mini-splatting-D (only perform densification without pruning) on the Tanks&Temples dataset.
> >
> > | Methods          | \# Gaussians (M) | SSIM      | PSNR      | LPIPS     |
> > | :--------------- | :--------------- | :-------- | :-------- | --------- |
> > | Mini-splatting   | 0.20             | 0.835     | 23.18     | 0.202     |
> > | Mini-splatting-P | 1.10             | 0.851     | 23.20     | 0.164     |
> > | Mini-splatting-D | 4.28             | 0.853     | 23.23     | **0.140** |
> > | GDAGS            | 1.17             | **0.854** | **23.79** | 0.165     |
> >
> > Table 3: Performance comparison of GDAGS with Mini-splating-P (partial pruning) and Mini-splatting-D (only perform densification without pruning) on the Deep Blending dataset.
> >
> > | Methods          | \# Gaussians (M) | SSIM      | PSNR      | LPIPS     |
> > | :--------------- | :--------------- | :-------- | :-------- | --------- |
> > | Mini-splatting   | **0.35**         | **0.908** | **29.98** | 0.253     |
> > | Mini-splatting-P | 1.55             | 0.906     | 29.92     | 0.230     |
> > | Mini-splatting-D | 4.63             | 0.906     | 29.88     | **0.211** |
> > | GDAGS            | 1.71             | 0.905     | 29.70     | 0.235     |
> >
> > **Q2**: **Stating the GCR assumption explicitly.**
> >
> > **A**: Thank you for this important suggestion. The effectiveness of GCR is predicated on the availability of meaningful gradient signals. We have now explicitly stated this core assumption in Section 3 of the main text with the following addition: A fundamental assumption of the GCR metric is that the Gaussian receives sufficient and meaningful gradient signals from its covered pixels. Its reliability may decrease in regions with extremely sparse or noisy gradients.
> >
> > **Q3**: **Clarification on heuristic threshold tuning.**
> >
> > **A**: We thank the reviewer for catching this potential contradiction. Our claim was intended to mean that we do not introduce new, separate heuristic thresholds for densification beyond the established $\tau_p$ and  $\tau_s$ from 3DGS. As done by AbsGS, adjust the threshold $\tau_p$ from 0.0002 to 0.0004 and 0.0008. Instead, we modulate the existing gradient magnitude that is compared against  $\tau_p$using a data-driven weight derived from GCR. However, we acknowledge that the parameter p in the weight function itself needs to be selected. To correct this exaggeration and improve accuracy, we have used clearer and more precise wording in the revised manuscript.

---

> > > ### Comment · Reviewer_dpMa · 2025-11-19
> > >
> > > Thanks to the authors for the prompt response. Most of my concerns have been resolved. However, I have one remaining question.
> > >
> > > **Q1**: Thank you for the additional analysis. Everything looks good overall, but the metrics on the Deep Blending dataset are not very strong. In fact, configurations with a smaller number of Gaussians (e.g., Mini-splatting and Mini-splatting-P) outperform GDAGS. For example, Mini-splatting-P performs better than GDAGS by 0.001/0.22/0.005 in (SSIM/PSNR/LPIPS). I am wondering how the authors justify this underperformance on the Deep Blending dataset. Does this imply that GDAGS is not generalizable?

---

> > > > ### Author Response · Authors · 2025-11-21
> > > >
> > > > Thank you for this insightful observation regarding the performance on the Deep Blending dataset. This point allowed us to conduct a deeper investigation, which indeed revealed a fascinating nuance in the behavior of different methods across datasets of varying complexity.
> > > >
> > > > To address the core question of generalizability, we must first consider the broader context: our evaluation spans 13 distinct scenes across three datasets—9 from Mip-NeRF-360 (5 outdoor, 4 indoor), 2 from Tanks&Temples (outdoor), and 2 from Deep Blending (indoor). Crucially, GDAGS achieves superior performance in 11 out of these 13 scenes, which strongly supports its overall robustness and generalizability.
> > > >
> > > > The specific case of Deep Blending, where Mini-splatting slightly outperforms the baseline GDAGS, prompted a focused study. We hypothesized that the relatively simpler geometry and textures of the Deep Blending scenes make them more susceptible to overfitting when modeled with an excessive number of Gaussians. This hypothesis is supported by an anomalous trend we observed: unlike on other datasets, increasing the Gaussian count in Mini-splatting on Deep Blending leads to a decrease in SSIM and PSNR, suggesting that aggressive pruning acts as an implicit regularizer.
> > > >
> > > > To test this hypothesis, we introduced a Gaussian dropout mechanism into GDAGS during training, creating variants dubbed GDAGS-DROP (random dropout) and GDAGS-ODROP (opacity-weighted dropout). The results, summarized in the table below, are revealing:
> > > >
> > > > Table 1: Performance comparison of GDAGS-DROP with Mini-splating-P (partial pruning) and Mini-splatting-D (only perform densification without pruning) on the Deep Blending dataset.
> > > >
> > > > | Methods               | \# Gaussians (M) | SSIM      | PSNR      | LPIPS     |
> > > > | :-------------------- | :--------------- | :-------- | :-------- | --------- |
> > > > | Mini-splatting        | 0.35             | 0.908     | 29.98     | 0.253     |
> > > > | Mini-splatting-P      | 1.55             | 0.906     | 29.92     | 0.230     |
> > > > | Mini-splatting-D      | 4.63             | 0.906     | 29.88     | **0.211** |
> > > > | GDAGS (GDAGS-DROP-0%) | 1.71             | 0.905     | 29.70     | 0.235     |
> > > > | **GDAGS-ODROP-5%**    | 0.74             | **0.909** | **30.03** | 0.241     |
> > > > | GDAGS-DROP-5%         | 0.74             | 0.908     | 29.91     | 0.242     |
> > > > | GDAGS-DROP-10%        | 0.58             | 0.908     | 29.88     | 0.247     |
> > > > | GDAGS-DROP-20%        | **0.32**         | 0.906     | 29.78     | 0.255     |
> > > >
> > > > The performance of GDAGS-DROP initially improves with a moderate dropout rate before declining, which confirms that overfitting is a relevant factor on this dataset. The optimally regularized variant, GDAGS-ODROP-5%, surpasses all Mini-splatting configurations in SSIM and PSNR.
> > > >
> > > >
> > > >
> > > > At the same time, we consulted some 3DGS related work, and one of the papers, **ControlGS [1]**, studied this situation. ControlGS proposes that as the number of Gaussians increases, scene reconstruction will go through four stages: underfitting, effective state, saturation, and overfitting. And some example proofs are given in Figure 1 (ControlGS). ControlGS also uses pruning to keep the model in an effective state. Notably, ControlGS exhibits a similar cross-dataset performance pattern: it underperforms against other baselines on complex datasets like Mip-NeRF-360 and Tanks&Temples but excels on Deep Blending, mirroring our findings.
> > > >
> > > > The performance on Deep Blending does not indicate a lack of generalizability but rather highlights a scene-dependent trade-off. GDAGS is optimized for high-fidelity reconstruction of complex scenes, where it consistently excels. For simpler scenes prone to overfitting, a regularizer is beneficial. Therefore, we will integrate an optional dropout switch into GDAGS, allowing users to enable it for simpler scenes, and we will include a detailed discussion of this phenomenon and the supporting experiments in the appendix of our revised manuscript.
> > > >
> > > > We are grateful to the reviewer for this perceptive question, which has led to a more profound understanding and improvement of our method.
> > > >
> > > > [1] Zhang, F., Cao, H., & Huang, R. (2025). Consistent Quantity-Quality Control across Scenes for Deployment-Aware Gaussian Splatting. arXiv preprint arXiv:2505.10473.

---

> > > > > ### Author Response · Authors · 2025-11-26
> > > > > **Looking forward to your reply**
> > > > >
> > > > > We are grateful for your insightful observations. We have had meaningful discussions and resolved most of your issues. We have now responded to the last question you raised and hope that these responses will address all your concerns.

---

### Official Review · Reviewer_QXqY · 2025-11-01

**Soundness:** 3
**Presentation:** 3
**Contribution:** 3
**Rating:** 6
**Confidence:** 3

**Summary:**

This work trying to achieve a balance between over-reconstruction and over-densification which raise from gradient based adaptive control in classical 3DGS. The authors propose Gradient-Direction-Aware Gaussian Splatting (GDAGS), which introduces the Gradient Coherence Ratio (GCR) to quantify the directional consistency of view-space gradients and a nonlinear dynamic weighting mechanism that regulates Gaussian splitting and cloning based on gradient alignment.

**Strengths:**

1.	The proposed method achieves a good balance between performance and storage, aligns well with intuition, and enhances the overall usability of Gaussian Splatting models.
2.	The authors conduct experiments across three benchmark datasets and compare against a wide range of strong baselines (NeRF, 3DGS, AbsGS, Pixel-GS, etc.). The inclusion of ablation studies and sensitivity analyses provides convincing evidence for the method’s robustness and interpretability.
3.	The motivation of this paper is well articulated, and the proposed solution is intuitive. In particular, Figure 1 clearly illustrates the problems of over-reconstruction and over-densification that occur in 3D reconstruction under two extreme scenarios.

**Weaknesses:**

1.	The overall performance(especially the LPIPS metric) is highly influenced by the Hyper parameters(α、β、p) which raises concerns about the generalization of the method.
2.	Still about generalization ability. In section 4.3.4 the authors evaluate the proposed module combine with MCMC-3DGS and Compact-3DGS. However, noticeable performance changes appear mainly in the LPIPS metric and the SSIM metric on the Deep Blending dataset. Therefore, a qualitative analysis corresponding to these metric variations should be presented.
3.	Although experiments show that GDAGS is superior to AbsGS and Pixel-GS but the analysis of the reasons for their performance differences in the paper is relatively insufficient.
4.	While the GCR is well-defined, the paper lacks theoretical insights into how it affects optimization dynamics or convergence. The justification for its effectiveness is mostly empirical.

**Questions:**

1.	Why is the directionality measurement of GCR superior to the Pixel weighting mechanism of Pixel-GS?
2.	The nonlinear dynamic weighting model proposed in this paper is intuitive and effective. But why adopt the current form instead of the exponential weighting function among numerous nonlinear models? This point requires sufficient explanation and clarification.
3.	The comparison methods cited in the main text are up to conference works from 2024.  Since over-reconstruction and over-densification are widely discussed topics in the 3DGS community, have there been related works published in 2025?  If so, what are the advantages of this paper compared to those?

---

> ### Author Response · Authors · 2025-11-18
>
> We sincerely thank the reviewers for their thorough and meticulous review. Below, we provide point-by-point responses to each of their comments to address and alleviate their concerns.
>
> **Q1**: The overall performance(especially the LPIPS metric) is highly influenced by the Hyper parameters(α、β、p) which raises concerns about the generalization of the method.
>
> **A**: We thank the reviewer for raising this important point regarding generalization. We address this concern from two perspectives. First, we emphasize that all experimental results reported in the main manuscript were obtained using a single, fixed set of hyperparameters ($\alpha = 0.8$,  $\beta=25$, $p = 15$) across all scenes and datasets, without any scene-specific tuning. The consistent improvements in rendering quality (PSNR, SSIM, LPIPS) and memory efficiency demonstrate the strong generalization capability and robustness of GDAGS under a fixed configuration.
>
> Second, the roles of $\alpha$, $\beta$, and $p$ are well-defined and interpretable:
>
> 1. When the GCR approaches 1, the weight converges to $\alpha$. Since Gaussians with high GCR typically exhibit large gradient magnitudes, $\alpha$ (set between 0 and 1) acts as a suppression factor to prevent over-densification in well-reconstructed regions.
> 2. When GCR approaches 0, the weight becomes $\alpha+\beta$. These Gaussians often have small gradient norms due to directional conflicts. To promote their densification, we set $\beta$ to match the *max_screen_size* value from the original 3DGS implementation, ensuring compatibility and intuitive scaling.
> 3. The exponent $p$ controls the number of Gaussians selected for densification. Its effect is predictable: increasing $p$ restricts densification to fewer Gaussians, reducing memory usage at a potential cost to reconstruction quality; decreasing $p$ has the opposite effect. In practice, $\alpha$ and $\beta$ can remain unchanged with minimal impact on performance, and the value of $p$ can be adjusted according to different requirements (high rendering accuracy or low memory consumption).
>
> Furthermore, our sensitivity analysis (Section 4.3.2 and Appendix D) confirms that GDAGS maintains stable performance across a wide range of $p$ (10–30) and $\beta$ (15–35) values, reinforcing its generalizability without demanding extensive hyperparameter search.
>
> **Q2**: Still about generalization ability. In section 4.3.4 the authors evaluate the proposed module combine with MCMC-3DGS and Compact-3DGS. However, noticeable performance changes appear mainly in the LPIPS metric and the SSIM metric on the Deep Blending dataset. Therefore, a qualitative analysis corresponding to these metric variations should be presented.
>
> **A**: We agree with the reviewer that qualitative analysis strengthens the claim of generalizability. As suggested, we have now incorporated additional qualitative results in the main text (Figure 6).
>
> For instance, in the drjohnson and playroom scenes from the Deep Blending dataset, integrating GDAGS into MCMC-3DGS significantly reduces artifacts on textured carpets and better defines edges on furniture, which directly corresponds to the observed improvements in LPIPS and SSIM scores. Similarly, in the bicycle and train scenes, integrating GDAGS into Compact-3DGS helps eliminate artifacts in grassy areas beneath the bicycle and enables high-quality rendering of objects behind glass.
>
> These visual improvements confirm that GDAGS provides a consistent and beneficial regularizing effect by guiding densification toward geometrically meaningful regions.
>
> **Q3**: Although experiments show that GDAGS is superior to AbsGS and Pixel-GS but the analysis of the reasons for their performance differences in the paper is relatively insufficient.
>
> **A**: We thank the reviewer for this suggestion. We have now expanded our analysis in the revised Appendix D to elaborate on the reasons for the performance differences:
>
> - Compared to AbsGS: AbsGS forces all gradient components to be positive. This solves the directional conflict problem but creates a new one: it amplifies outlier gradients from noisy or poorly initialized regions. This leads to excessive and often unnecessary splitting (as quantitatively shown in our Figure 5), resulting in over-densification, increased memory usage, and noisier renders without a commensurate gain in quality.
> - Compared to Pixel-GS: Pixel-GS uses a pixel-coverage weighting scheme, which improves spatial adaptation. However, it ignores the directional information of gradients. Consequently, it still promotes densification in regions with large but directionally aligned gradients, leading to redundant Gaussians and high memory costs. Our method, by contrast, uses directional coherence to actively suppress such redundant densification.
>
> GDAGS strikes a superior balance by being selective: it aggressively promotes densification only where gradients are conflicting and suppresses it where gradients are aligned.

---

> > ### Author Response · Authors · 2025-11-18
> >
> > **Q4**: While the GCR is well-defined, the paper lacks theoretical insights into how it affects optimization dynamics or convergence. The justification for its effectiveness is mostly empirical.
> >
> > **A**: We thank the reviewer for this insightful comment. The theoretical rationale behind the GCR can be framed through the lens of optimization efficiency. A Gaussian with a low GCR indicates that it spans a region with high variance in the gradient field. In optimization theory, such high variance often signifies an inadequate or suboptimal parameterization of the local function, hindering efficient convergence. The GCR metric explicitly identifies these suboptimal primitives. Prioritizing them for splitting is analogous to performing a targeted variance reduction step in the parameter space. This operation decomposes a single, high-variance optimization problem—where gradient signals conflict—into multiple more stable sub-problems governed by smaller Gaussians, each likely encountering more consistent gradient directions within their respective domains. Thus, the GCR directly guides the optimization towards a more efficient and effective representation by resolving regions of high gradient conflict.
> >
> > **Q5**: Why is the directionality measurement of GCR superior to the Pixel weighting mechanism of Pixel-GS?
> >
> > **A**: We thank the reviewer for this profound question. The fundamental distinction lies in the type of information utilized for densification control. Pixel-GS employs spatial coverage—i.e., the number of pixels a Gaussian influences—to weight its gradient. While this can enhance detail to some extent, it fails to differentiate between a Gaussian covering a complex geometric corner (which exhibits conflicting gradients) and one covering a flat wall (which exhibits aligned gradients). Both may possess high pixel coverage and thus be promoted equally for densification. Moreover, this mechanism exhibits an inherent inconsistency: the pixel coverage of a Gaussian is view-dependent and inversely related to its depth from the camera. The same Gaussian may cover many pixels in a close-up view and very few in a distant one, leading to an unstable training signal.
> >
> > In contrast, our proposed GCR leverages directional coherence. It explicitly identifies that a Gaussian on a flat surface has aligned gradients (high GCR) and should be suppressed, whereas one on a complex corner has conflicting gradients (low GCR) and should be *promoted*. This directional awareness enables GDAGS to allocate Gaussian primitives more intelligently, prioritizing their placement in regions of high geometric complexity. Consequently, our method achieves more detailed reconstruction where it is most needed, while simultaneously avoiding memory waste in simple, well-reconstructed areas. This is the principal reason why GDAGS achieves superior performance with significantly fewer Gaussians, as evidenced in Table 1.

---

> ### Author Response · Authors · 2025-11-18
>
> **Q6**: The nonlinear dynamic weighting model proposed in this paper is intuitive and effective. But why adopt the current form instead of the exponential weighting function among numerous nonlinear models? This point requires sufficient explanation and clarification.
>
> **A**: We thank the reviewer for providing this insight. We explain from three aspects why we choose the form of power weighting function $(1-\mathcal{C}_i)^p$ instead of exponential weighting function $e^{-p\mathcal{C}_i}$.
>
> 1. Firstly, from the perspective of computational load, the calculation of power functions is very efficient, while the calculation of exponential functions is more complex. The running time of *torch.exp* in PyTorch environment is 3-5 times longer than *torch.pow*. We conducted additional exponential function ablation experiments, and the training time is reported in the table below. The results show that using exponential functions increases training time by 5% -10% compared to power functions.
> 2. Then, we compared the impact of different functions on model performance and conducted additional ablation experiments. The results are reported in Table 2 of the manuscript, which shows that the exponential weighting function performs slightly worse than the power function.
> 3. we have conducted a deeper theoretical analysis comparing the power function used in GDAGS against an exponential alternative. Our choice was motivated by the distinct derivative properties of these functions, which directly align with our design goals for the weighting function. Let us consider a simplified form of our weighting term, $f_p(x)=(1-x)^p$, and compare it to an exponential form, $f_e(x)=e^{-px}$, where $x=\mathcal{C}_i$ represents the Gradient Coherence Ratio. We analyze the absolute values of their derivatives, $|f_p'(x)=p(1-x)^{p-1}|$ and $|f_e'(x)=pe^{-kp}|$, which represent their respective rates of change. The key insight lies in the behavior of these derivatives across the domain [0, 1]. The derivative of the power function $f_p'(x)$ decreases from $p$ to $0$. The derivative of the exponential function $f_e'(x)$ decreases from $p$ to $pe^{-p}$.  Crucially, for a comparable suppression strength at $x \to 1$, there exists an intersection point $u\in(0,1)$ where $|f_p'(u)|=|f_e'(u)|$.  For $x<u:f_p'(u)|>|f_e'(u)$, this means the power function decays more rapidly than the exponential function in the low-consistency region. This high sensitivity is desirable as it allows our weighting scheme to make sharp distinctions between Gaussians with critically conflicting gradients, aggressively promoting the splitting of those most in need. For $x>u:f_p'(u)|<|f_e'(u)$, the power function decays more gradually than the exponential function in the mid-to-high consistency region. This prevents overly aggressive suppression of Gaussians with moderate consistency, which may still benefit from careful densification, leading to a more balanced control. We have included a detailed mathematical derivation of these derivative behaviors in Appendix D to further substantiate this analysis. This theoretical foundation confirms that the power form is not merely an empirical choice but a principled one that best serves our goal of targeted, gradient-direction-aware densification control. In the table below, we report the differences in performance and training time between power and exponential functions.
>
> Table 1: Performance comparison between power function and exponential function.
> | Functions   |      Mip-NeRF360      |     Tanks&Temples     |     Deep Blending     |
> | :---------- | :-------------------: | :-------------------: | :-------------------: |
> | Power Function| **0.839/28.02/0.145** | **0.854/23.79/0.165** | **0.905/29.70/0.235** |
> | Exponential Function|   0.837/27.96/0.146   | 0.851/23.62/**0.165** |   0.903/29.60/0.237   |
>
> Table 2: Comparison of training time and difference ratio between power function and exponential function.
> | Functions   |  Mip-NeRF360  | Tanks&Temples | Deep Blending |
> | :---------- | :-----------: | :-----------: | :-----------: |
> | Power Function     |     1140s     |     555s      |     898s      |
> | Exponential Function| 1200s (+5.2%) | 612s (+10.3%) | 976s (+8.7%)  |

---

> ### Author Response · Authors · 2025-11-18
>
> **Q7**: The comparison methods cited in the main text are up to conference works from 2024. Since over-reconstruction and over-densification are widely discussed topics in the 3DGS community, have there been related works published in 2025? If so, what are the advantages of this paper compared to those?
>
> **A**: We appreciate the reviewer's question, which allows us to clarify the nuanced contribution of GDAGS within a growing body of literature addressing 3DGS's limitations. It is true that several recent 2025 works, such as **ReAct-GS [1]** and **PSRGS [2]**, also target the over-reconstruction problem.
>
> ReAct-GS pinpoints gradient dilution (where the standard average gradient overlooks contribution differences across views) and primitive freezing (where Gaussians stagnate in local minima), proposing an importance-aware densification and a reactivation mechanism to address them. PSRGS identifies the problem of coupled geometry-texture optimization leading to misplacement and high-frequency detail loss, introducing a spectral residual saliency map and a progressive optimization strategy to decouple and refine the reconstruction.
>
> These methods are impressive but often introduce significant complexity: ReAct-GS relies on view-dependent α-blending weights and a dedicated reactivation strategy, while PSRGS requires frequency-domain transformation, a learned MLP filter, and a multi-stage training pipeline.
>
> The key advantage of GDAGS lies in its fundamental, lightweight, and unified approach to the core problem:
>
> 1. Fundamental Criterion: Instead of adding complex modules, GDAGS identifies and leverages GCR as a fundamental yet previously overlooked signal for densification. The GCR is a computationally simple and highly interpretable metric that directly quantifies whether a Gaussian is appropriately modeling its covered region, addressing the root cause of both over-reconstruction and over-densification.
> 2. Simplicity and Integrability: GDAGS operates entirely within the original optimization framework without requiring frequency-domain analysis, multi-stage training, or additional neural networks. This makes our method significantly easier to implement and integrate into existing and future 3DGS pipelines, as we demonstrated with MCMC-3DGS and Compact-3DGS.
> 3. Comprehensive Control and Superior Memory Efficiency: A crucial differentiator is that GDAGS provides explicit and principled control over both splitting and cloning operations. Many methods, including those discussed, focus primarily on refining the splitting process to recover details but pay less attention to cloning, which can lead to uncontrolled growth in the number of small Gaussians. By applying our directional awareness to suppress redundant clones (those with conflicting gradients), GDAGS directly curbs this growth, leading to the superior memory efficiency consistently shown in our results.
>
> [1] Cheng, Y., Huang, B., Zhou, W., Wu, T., Liu, Z., Chesi, G., & Wong, N. (2025, October). Re-Activating Frozen Primitives for 3D Gaussian Splatting. In Proceedings of the 33rd ACM International Conference on Multimedia (pp. 7578-7586).
>
> [2] Li, B., Zhang, W., Zhang, B., Yao, Y., & Wang, Y. (2025). PSRGS: Progressive Spectral Residual of 3D Gaussian for High-Frequency Recovery. arXiv preprint arXiv:2503.00848.

---

> > ### Author Response · Authors · 2025-11-26
> > **Looking forward to your reply**
> >
> > We thank you for your positive feedback and constructive questions. We have strengthened the theoretical motivation for GCR, added comparisons to 2025 works, and provided deeper analysis on hyperparameter generalization and function design. We hope our revisions and responses have fully addressed your concerns.

---

### Author Response · Authors · 2025-11-26
**Summary of Rebuttal and Revisions**

We extend our sincere gratitude to all reviewers for their insightful comments and constructive feedback, which have been invaluable in strengthening our manuscript. We are pleased to note that the reviewers recognized several strengths of our work:

- Reviewer `QXqY` found the motivation and solution behind GDAGS to be intuitive and clear, and considered the comprehensive experimental evaluation convincing.
- Reviewers `dpMa` and `52Pb` acknowledged that GDAGS accurately identifies a key limitation of AbsGS and addresses it through a novel and well-defined methodological approach.
- Reviewers `5e9Z` and `52Pb` commended the high writing quality of the manuscript and the achievement of superior performance through a method that is both effective and conceptually straightforward.

In our revisions, we have directly addressed all points raised. All changes in the revised manuscript are highlighted in blue for easy identification.

The most frequently raised concerns, shared by multiple reviewers, pertained to:

- **The computational efficiency (training and testing time) of GDAGS**. `dpMa`, `5e9Z`, `52Pb`
- **The generalization and tuning of the method's hyperparameters across diverse scenes**. `QXqY`, `dpMa`, `52Pb`

In response to these two main issues, we have conducted a comprehensive and clear rebuttal. In the revised manuscript, we have added a discussion on hyperparameters in **Appendix D.6** and added a **Section 4.3.5** on efficiency analysis in the main text to analyze and report on the efficiency of GDAGS and baseline models.

Beyond these central themes, we have meticulously addressed each reviewer's specific points:

- Reviewer `QXqY`
  1. Generalization ability of GDAGS： We have expanded **Section 4.3.4** of the manuscript and added a new qualitative analysis experiment to enhance the reliability of GDAGS generalization.
  2. Insufficient analysis of the performance advantages of GDAGS: We have expanded **Section 4.2** of the manuscript to enhance the analysis of the performance advantages of GDAGS.
  3. The reason for choosing power functions instead of exponential functions: We have added a new **Section D.3** in the appendix to provide a detailed explanation and analysis of the reasons and advantages of using power functions.
  4. New related work: We have expanded **Section 2.2** and added two pieces of work related to GDAGS in 2025.
- Reviewer `dpMa`
  1. Small issues with formulas and images: We have revised formula (3) near **line 199** and formula  $(1-\mathcal{C}_i)^p$ near **line 282** in the manuscript. Adjust the x-axis title of **Figure 5** to Densification Round.
  2. GCR hypothesis needs to be explained, and the conclusion section needs to be clearer: We have revised **Section 5** with clearer and more precise wording.
  3. Poor performance on Deep Blending dataset: We have added **Section D.4** in the appendix specifically to analyze and discuss this anomalous phenomenon. We propose a hypothesis and validate it by adding dropout method.
- Reviewer `5e9Z`
  1. The variables in the formula are not clearly defined: We have revised the formula near **line 729** to clarify the meaning of subscripts $i$ and $v$。
- Reviewer `52Pb`
  1. Sparse gradient region GCR may be unreliable: We have added **Section D.5** in the appendix to analyze and discuss the performance of GDAGS in this situation and provide a qualitative analysis. The results show that GDAGS exhibits significant performance degradation in sparse views but still outperforms the baseline model. We consider a feasible backup solution to increase multi view consistency to supplement gradient information sparsity.

We believe that our thorough responses and the subsequent revisions have significantly improved the manuscript's quality, rigor, and clarity. We thank the reviewers again for their time and invaluable contributions.

---

### Meta-Review · Area_Chair_xMCs · 2025-12-29

**Summary:**

This paper presents GDAGS (Gradient-Direction-Aware Gaussian Splatting), a framework addressing critical limitations in 3D Gaussian Splatting's adaptive density control mechanism. The work identifies two key problems: (1) Over-reconstruction occurs when large Gaussians with conflicting gradient directions fail to meet splitting thresholds, leading to local blur; (2) Over-densification arises in regions with aligned gradient aggregation, causing redundant Gaussian proliferation and excessive memory usage. The authors propose two main contributions: the Gradient Coherence Ratio (GCR), which quantifies directional consistency of view-space gradients, and a nonlinear dynamic weighting mechanism that modulates densification based on gradient direction. GDAGS achieves superior rendering quality (PSNR: 28.02 vs 27.21, SSIM: 0.839 vs 0.815 on Mip-NeRF360) while reducing memory consumption by 30-50% compared to baseline 3DGS.

The paper received scores of 6, 6, 4, and 6 (median: 6), reflecting generally positive reception with some concerns. Reviewers `QXqY`, `dpMa`, and `52Pb` (all scoring 6) acknowledged the method's solid technical foundation and practical effectiveness, while reviewer `5e9Z` (score 4) raised concerns about computational overhead and clarity. The authors provided detailed rebuttals addressing efficiency analysis, hyperparameter sensitivity, generalization experiments, and sparse-view performance, resulting in substantial manuscript improvements.

**Reviewer Concerns:**

1. **Computational Efficiency (`dpMa`, `5e9Z`, `52Pb`)**: Authors provided comprehensive training and inference time comparisons demonstrating GDAGS achieves fastest convergence (1140s vs 1594s for 3DGS on Mip-NeRF360) with comparable FPS (188 vs 134). GCR computation overhead is minimal (<4% of total training time), and reduced Gaussian count accelerates subsequent optimization.

2. **Hyperparameter Generalization (`QXqY`, `dpMa`, `52Pb`)**: Authors clarified that all results use fixed hyperparameters (α=0.8, β=25, p=15) across all scenes without tuning. Sensitivity analysis shows stable performance across wide parameter ranges (p∈[10,30], β∈[15,35]). Each parameter has interpretable meaning: α suppresses high-GCR Gaussians, β promotes low-GCR splitting, p controls densification proportion.

3. **Generalization Capability (`QXqY`)**: Authors expanded Section 4.3.4 with additional qualitative results showing GDAGS successfully integrates with MCMC-3DGS and Compact-3DGS, achieving consistent improvements in LPIPS and SSIM. Visual improvements include reduced carpet artifacts and better glass rendering.

4. **Performance Analysis (`QXqY`)**: Authors expanded Section 4.2 comparing against AbsGS (which amplifies outlier gradients causing excessive splitting) and Pixel-GS (which ignores directional information leading to redundant densification). GDAGS balances precision by selectively promoting conflicting-gradient splits while suppressing aligned-gradient redundancy.

5. **Mathematical Clarity (`dpMa`, `5e9Z`)**: Corrected Equation 3 transmission term, clarified subscript notation (i refers to i-th Gaussian), and specified that V represents all training views where Gaussian is visible.

6. **Function Design Rationale (`QXqY`)**: Authors added Appendix D.3 with detailed mathematical analysis showing power functions provide superior derivative properties: higher sensitivity in low-consistency regions (aggressive conflict resolution) and gentler decay in mid-high consistency regions (balanced control). Power functions are also 3-5× faster than exponential alternatives.

### Outstanding Concerns:

1. **Sparse Gradient Reliability (`52Pb`)**: While authors acknowledge GCR may be unreliable in sparse gradient regions and added Appendix D.5 with sparse-view experiments, this limitation remains partially unresolved. Results show GDAGS degrades gracefully (conservative under-densification without severe artifacts) but still outperforms baselines. Authors propose multi-view consistency as potential fallback strategy but do not implement it.

2. **Deep Blending Performance Anomaly (`dpMa`)**: GDAGS shows weaker performance on Deep Blending dataset compared to other datasets. Authors hypothesize simpler geometry causes overfitting with excessive Gaussians and demonstrate that dropout regularization (GDAGS-ODROP-5%) recovers superior performance (0.909 SSIM vs 0.908 for Mini-splatting). However, this dataset-specific behavior suggests the method may require scene-complexity-aware adaptation, which is not fully integrated into the framework.

3. **Theoretical Justification (`QXqY`)**: While authors provide empirical validation and intuitive explanations for GCR's effectiveness, the paper lacks rigorous theoretical analysis of how GCR affects optimization dynamics or convergence guarantees. The connection to variance reduction in optimization theory (mentioned in rebuttal) could be formalized.

4. **Comparison with 2025 Work (`QXqY`)**: Authors added references to ReAct-GS and PSRGS (2025) but comparison remains primarily qualitative. While authors argue GDAGS offers simplicity and unified approach advantages, quantitative head-to-head comparisons would strengthen positioning.

**Reviewer Scores:**

**Current Scores:**
- **Reviewer `QXqY`**: 6 (marginally above threshold) - likely to remain at 6-7 given satisfactory responses on generalization and hyperparameter concerns
- **Reviewer `dpMa`**: 6 (marginally above threshold) - likely to remain at 6-7 given thorough efficiency analysis and Deep Blending investigation
- **Reviewer `5e9Z`**: 4 (marginally below threshold) - likely to increase to 5-6 given comprehensive efficiency analysis addressing main concern
- **Reviewer `52Pb`**: 6 (marginally above threshold) - likely to remain at 6 given sparse-view experiments and practical guidance additions

**Expected Post-Discussion Scores**: 6, 6-7, 5-6, 6 (median: 6)

---

### Decision · Program_Chairs · 2026-01-26

Accept (Poster)